# CryoEM structures of the multimeric secreted NS1, a major factor for dengue hemorrhagic fever

Bo Shu[1,2], Justin S. G. Ooi [1,2], Aaron W. K. Tan [1,2], Thiam-Seng Ng [1,2], Wanwisa Dejnirattisai[3], Juthathip Mongkolsapaya [3], Guntur Fibriansah [1,2], Jian Shi[2], Victor A. Kostyuchenko[1,2], Gavin R. Screaton [3] & Shee-Mei Lok [1,2]

Dengue virus infection can cause dengue hemorrhagic fever (DHF). Dengue NS1 is multifunctional. The intracellular dimeric NS1 (iNS1) forms part of the viral replication complex. Previous studies suggest the extracellular secreted NS1 (sNS1), which is a major factor contributing to DHF, exists as hexamers. The structure of the iNS1 is well-characterised but not that of sNS1. Here we show by cryoEM that the recombinant sNS1 exists in multiple oligomeric states: the tetrameric (stable and loose conformation) and hexameric structures. Stability of the stable and loose tetramers is determined by the conformation of their N-terminal domain – elongated β-sheet or β-roll. Binding of an anti-NS1 Fab breaks the loose tetrameric and hexameric sNS1 into dimers, whereas the stable tetramer remains largely unbound. Our results show detailed quaternary organization of different oligomeric states of sNS1 and will contribute towards the design of dengue therapeutics.

Dengue is a positive sense single-stranded RNA enveloped virus. It belongs to the family *Flaviviridae*, which also includes significant human pathogens, such as West Nile virus (WNV), yellow fever virus (YF) and Zika virus (ZIKV). Dengue virus causes disease that ranges from mild dengue fever to the severe dengue hemorrhagic fever (DHF) and dengue shock syndrome (DSS)[1]. There are four DENV serotypes. Dengvaxia, which is currently the only licensed dengue vaccine, has an overall efficacy of only 45%, and there is a possibility of priming the previously immune naïve children to develop severe disease in subsequent natural infection after vaccination[2]. It has been proven to be difficult to develop an effective vaccine using whole dengue virus or its surface E protein, as the vaccine would need to stimulate equally strong immune responses simultaneously against all four dengue serotypes or else it can lead to a possibility of antibody-dependent enhancement (ADE) of infection[3]. This could lead to severe DHF/DSS.

Recently, dengue non-structural protein 1 (NS1) has been investigated as an alternative protein candidate for vaccine development. It has been shown to play an important role in causing the vascular leakage symptoms in DHF/DSS[4,5]. NS1 is secreted alongside dengue virus particles during an infection. It is a multifunctional protein which exists in different oligomeric forms intracellularly and extracellularly[6], and they play distinctly different roles. The intracellular NS1 (iNS1), exists as a membrane associated dimeric form and is a part of the viral replication complex. The extracellular secreted NS1 (sNS1), exists in higher oligomeric forms. sNS1 likely causes vascular leakage symptoms in DHF/DSS via two pathways—indirectly, by stimulating the innate immune response through activation of complement and Toll-like receptors[7], and directly, by binding to endothelial cells causing endothelial hyperpermeability. sNS1 when bound directly to endothelial cells causes barrier dysfunction of the cell monolayer as demonstrated in an in vitro *trans*-endothelial

---

[1]Programme in Emerging Infectious Diseases, Duke-National University of Singapore Medical School, Singapore 169857, Singapore. [2]Centre for Bioimaging Sciences, Department of Biological Sciences, National University of Singapore, Singapore 117557, Singapore. [3]Medical Sciences Division, University of Oxford, Oxford OX3 9D, UK. e-mail: sheemei.lok@duke-nus.edu.sg

electrical resistance (TEER) assay[4,8]. Mice injected with sNS1 along with a sublethal dose of DENV were observed to succumb to infection. Injection of sNS1 alone also causes vascular leakage in mice[4]. NS1-immune polyclonal serum and anti-NS1 monoclonal antibodies (MAbs) can protect mice against lethal DENV2 challenge[4]. sNS1 was also found to enhance viral infection in mosquitoes by down-regulating mosquito midgut immune genes[9]. sNS1 is therefore an important viral protein both for viral infection (mammalian cells and mosquitoes) and pathogenesis, and it is important to understand the structure of the sNS1 protein.

iNS1 (45 kDa) is first made in the endoplasmic reticulum (ER) as a monomer. It forms dimers in the ER and then is glycosylated in the trans-Golgi network. Some NS1 is transported to the cell membrane[10], while others are secreted outside the cell, becoming the sNS1. There is a high-resolution crystal structure of full length dimeric dengue iNS1[11] produced by baculovirus expression and subsequent extraction from cellular membrane compartments. There are also crystal structures of the dimeric form of NS1 protein from other flaviviruses[12] and they are largely similar to that of the DENV iNS1. Each protomer of the iNS1 contains three domains (Supplementary Fig. 1a): (1) the N-terminus contains the β-roll domain (amino acids 1–30) that interacts with the same domain from the other protomer within the dimer to form a roll-like structure (named β-roll). This domain is generally hydrophobic and is thought to interact with cellular membranes. (2) Following the β-roll domain is the wing domain (amino acids 31–181). The wing domain is the most flexible domain as it differs between different flavivirus NS1 crystal structures[12]. (3) the β-ladder domain (amino acids 182–352), consists of a continuous arrangement of 10 β-strands. The β-ladder domains from the two iNS1 protomers line up with each other to form a long ladder-like pattern. One side of the β-ladder faces the hydrophobic β-roll, while the other side is decorated with its highly charged spaghetti loops. The interaction between protomers within the dimer is stabilized by a large number of hydrogen bonds[13].

About a decade ago, two low-resolution (23 Å and 30 Å) EM structures of sNS1 were determined: a cryoEM structure reconstructed from images acquired by using a negative-stain room temperature EM structure (Supplementary Fig. 1b)[14] and the other, a 120 kV LaB₆-equipped electron microscope (Philip CM12 TEM)[15]. Both maps had an overall hollow barrel-like structure lacking fine structural details. Since then, the availability of 300 kV transmission electron microscopes equipped with direct electron detectors and field emission gun electron sources, as well as improved cryoEM reconstruction methods, have made more routine determination of cryoEM structures of small proteins to high-resolution possible.

Here we show the cryoEM structures of the recombinant sNS1 tetramers (dimer of dimers) and hexamers (trimer of dimers) from a DENV2 clinical strain PVP94/07. We observe sNS1 exists predominantly in the tetrameric form. There are two tetrameric states—stable and loose. The stable tetramer structure is determined to ~3.5 Å resolution while the overall resolution of the loose tetramer is 8 Å. We have also done focused refinement on one of the dimers of the loose tetramer and it is determined to 3.4 Å resolution. The most obvious and dramatic difference between the stable and loose tetramers is the organization of their N-terminal domains—elongated β-sheet and β-roll, respectively. Only a minority of the recombinant sNS1 population is hexameric and this cryoEM structure was determined to 8 Å resolution. We also determined a structure of sNS1 complexed with a Fab 5E3 to 3.5 Å resolution. Most of the sNS1 (loose tetramers and hexamers) dissociates into dimers when bound to Fab, while the stable tetramer remains largely unbound. This is consistent with our results showing the ability of the Fab 5E3 to only partially inhibit sNS1-induced endothelial cell permeability. This study shows the recombinant sNS1 exists in a heterogenous mixture of oligomerization states, and we present the high-resolution structures of the recombinant sNS1.

## Results

### The recombinant sNS1 from DENV2 contains heterogenous oligomeric states with tetramers as the predominant population

Recombinant NS1 with His-tag at its C-terminus (DENV2 clinical strain PVP94/07) was expressed in human embryonic kidney cells. We performed crosslinking with bis(sulfosuccinimidyl) suberate (BS³) before analyzing it by denaturing SDS-PAGE. This showed the presence of aggregates, which could be due to BS³ nonspecifically crosslinking proteins, as well as bands for hexamer, tetramer, dimer, and monomer (Fig. 1a). For comparison, we also harvested sNS1 from DENV2 PVP94/07 infected cells and conducted a similar crosslink assay. The sNS1 from infected cells (Fig. 1b) without BS³ incubation (0 min) was found to form tetramers, dimers and monomer that is different from our recombinant protein, while more tetramers were observed with 10 and 20 min incubation.

Sedimentation velocity analytical ultracentrifugation (AUC) experiment of the recombinant sNS1 was conducted, and the molecular weights (MW) were derived by data analysis with SEDFIT based on sedimentation coefficient (Supplementary Fig 2). The estimated MW of the first peak is 32.5 kDa, likely corresponds to a very minute population of monomeric particles. The MWs of the second and third peaks are 146 and 251 kDa, and they correspond to tetrameric and hexameric sNS1 particles, respectively. The tetramers form the largest population of the sNS1 particles.

We collected 7714 cryoEM micrographs (Supplementary Table 1) and observed particles with distinct shapes (Fig. 1c). We then performed several cycles of 2D classification and alignment of the boxed particles. The 2D class averages show particles with different features (Fig. 1d)—some appearing like stable and loose tetramers and others like hexamers. We did not observe any sNS1 dimers. This suggests that sNS1 consists of heterogenous populations of particles with likely tetrameric and hexameric states.

### CryoEM reconstruction of different oligomeric states of the recombinant sNS1

As the recombinant sNS1 consists of a heterogenous population of particles of different oligomerization states, we performed a 3D classification using the program Relion[16] (Supplementary Fig. 3). The first round of unsupervised classification was done assuming no symmetry (C1) to avoid bias towards any specific oligomerization states. We then performed rough fitting with a dimeric NS1 crystal structure into these maps to determine their oligomerization states. We observed two maps that correspond to tetramers: one with clear densities suggesting that the overall structure is quite stable, while the other with looser interactions between the two dimers and poorer map resolution (Supplementary Fig. 3). We refer the former as stable tetramer and the latter as loose tetramer. We also observed a 3D class that is hexameric in structure (Supplementary Fig. 3). Consistent with the 2D class averages, the 3D classification also did not yield any dimeric structures. We then used the maps from 3D classification in Relion as initial models for further classification and refinement in the program CryoSPARC[17]. For the stable tetramers, we performed multiple cycles of heterogenous and non-uniform refinement[18] by imposing D2 symmetry and obtained a final map of 3.5 Å resolution (Fig. 2a and Supplementary Fig. 4), as measured by gold standard FSC curve cutoff at 0.143. For the loose tetramers, we performed heterogenous and non-uniform refinement with no symmetry imposed, and we determined one subclass of the loose tetramer to 8 Å resolution (Fig. 3a (i) and Supplementary Fig. 5a). We then performed local refinement on one of its dimers and obtained a 3.4 Å cryoEM density map (Fig. 3a (ii) and Supplementary Fig. 5b). For the hexamers, since we only had a limited number of particles (~30,000 particles), we performed only non-uniform refinement with D3 symmetry imposed and obtained an 8 Å map (Fig. 3b and Supplementary Fig. 6). The cryoEM reconstruction flow chart is

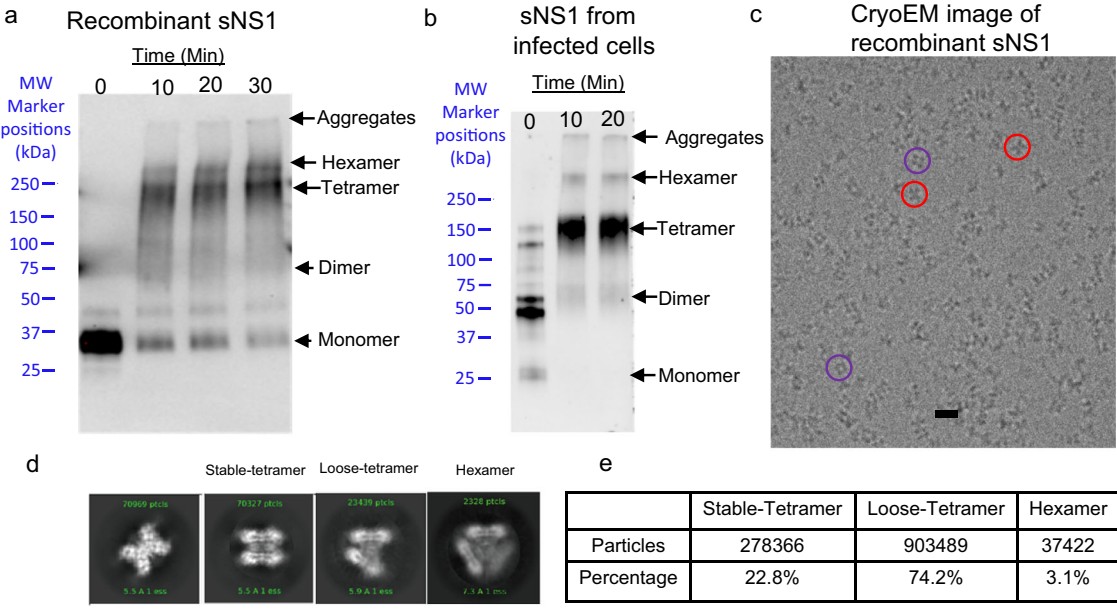

**Fig. 1 | Profile of the recombinant sNS1 and sNS1 from the supernatant of infected cells of DENV2 (PVP94/07) clinical strain.** Western blot denaturing SDS-PAGE gel of sNS1 crosslinked with BS³ at different incubation times: **a** recombinant C-terminal His-tag sNS1 (detected using Anti-polyHistidine−Peroxidase antibody) and **b** the sNS1 from the supernatant of infected cells (detected using anti-NS1 antibody). For the recombinant sNS1, aggregates, hexamers, tetramers, dimers and monomers were seen. For the sNS1 of infected cells crosslinked sample, we observed aggregates, hexamers, tetramers and dimers. All samples were added the SDS-PAGE loading buffer, boiled for 5 min and then analyzed on a 4−20% SDS-PAGE gradient gels. **c** A cryoEM micrograph of the recombinant sNS1 showing distinct-shaped particles, e.g., butterfly-shaped (red circles) and cube-shaped particles (purple circles). Scale bar is 10 nm. For **a**, **b**, and **c**, we have repeated the experiments at least 2 times. **d** The 2D class averages indicate that the particles are heterogenous, with some likely existing as tetramers (stable and loose forms) and others as hexamers. The first panel represents the typical 2D average side view which is similar between all oligomeric states and the other three represent the top views of different oligomeric forms. **e** The distribution of particles in different oligomerization states determined after 3D classification and refinement. Most of the particles are tetramers (loose and stable) and hexamers are a minority in the population. Source data are available as a Source Data file.

summarized in Supplementary Fig. 3 and validation statistics in Supplementary Table 1.

## Structure of the recombinant sNS1 stable tetramers

The 3.5 Å resolution recombinant sNS1 stable tetramer cryoEM map shows clear separation between β-strands (Supplementary Fig. 4c, d). We also calculated the local resolution (Supplementary Fig. 4a) of the map. While most of the density map is ~3.5 Å resolution, parts of the map (corresponding to the β-ladder) are at ~3.0 Å resolution with some of the side chain densities resolved (Supplementary Fig. 4d). The resolution of the wing domain and the β-roll was lower (4.0–4.5 Å). The density of both N- (residues 1–13) and C-terminal ends (residues 341–352) is missing from the map.

We interpreted the stable tetramer (dimer of dimers) cryoEM density by fitting in the zika dimeric NS1 crystal structure (PDB:5GS6), because the structure is more complete than the dengue iNS1, and then mutated it into dengue NS1 sequence (Fig. 2a, b and Supplementary Fig. 4b). After fitting, we compared one of our sNS1 dimers in the stable tetramer to the dengue iNS1 dimer (PDB:4O6B) and it shows the wing domain of our sNS1 dimer is rotated by 6.8° with respect to the β-ladder. The β-roll structures of the two dengue iNS1 dimers cannot fit well into the cryoEM density of the stable tetramer and they clash with each other. Indeed, the space (25 Å) between two dimers within the stable tetramers is not big enough to accommodate the two iNS1 β-rolls. We observed that the β-roll densities, instead of concentrating at the middle of the dimers as observed in iNS1 (Supplementary Fig. 1), have an elongated shape that stretches across the two protomers within a dimer (Fig. 2a, b). This suggests that they have very different tertiary structures. Fitting of the density map shows that residues 18–27 form a 27 Å long β-strand which pairs with the same β-strand of the other protomer within the dimer to form a long antiparallel β-sheet (hereafter, referred to as elongated β-sheet) (Fig. 2a, b,

Supplementary Movie 1). As the resolution of this region is ~4.5 Å, we analyzed their interaction by using a distance cutoff of 8 Å between the Cα backbone. Using this criterion, we observed that this elongated β-sheet from one dimer likely interacts with the same β-sheet from the opposite dimer via electrostatic interactions or hydrogen bonds at both ends of each respective β-sheet (Fig. 2c (i)−(ii)). Charge analysis in UCSF Chimera[19] shows that these two ends have high charge complementarity between dimers (Fig. 2c (iii)), suggesting possible hydrogen bonds or electrostatic interactions. Hydrophobic analysis shows that these β-sheets have an overall highly hydrophobic surface that also plays an important role in stabilizing their interactions.

We next compared the β-roll domain of iNS1 with the elongated β-sheet of the sNS1 stable tetramer. The iNS1 β-roll domain contains three β-strands (residues 1–7, 12–16, and 18–22), whereas we were unable to observe any density for residues 1–13 in the sNS1 stable tetramer, probably due to disorder in this region, and residues 18–27 form a long, continuous β-strand. This raises a question: is the sNS1 tetramer assembled from dimers of iNS1? If so, is it possible for the iNS1 β-roll structure to rearrange into the elongated β-sheet seen in the stable tetramer? If this were to happen, it would require the second β-strand of iNS1 to rotate by ~120° with respect to the third β-strand to form the long, continuous β-strand we observed (Fig. 2d). Whether the iNS1 dimer is a precursor of the sNS1 stable tetramer is unknown. It is also possible that the sNS1 stable tetramer is assembled during translation of the NS1 protein by a pathway independent of iNS1 dimer formation, before being secreted.

## CryoEM structure of the recombinant loose sNS1 tetramers

We also determined an 8 Å resolution structure of the recombinant loose sNS1 tetramer (Fig. 3a (i) and Supplementary Fig. 5a), and by focused refinement on one of its constituent dimers, we obtained a 3.4 Å resolution cryoEM map (Fig. 3a (ii) and Supplementary Fig. 5b (i–ii)).

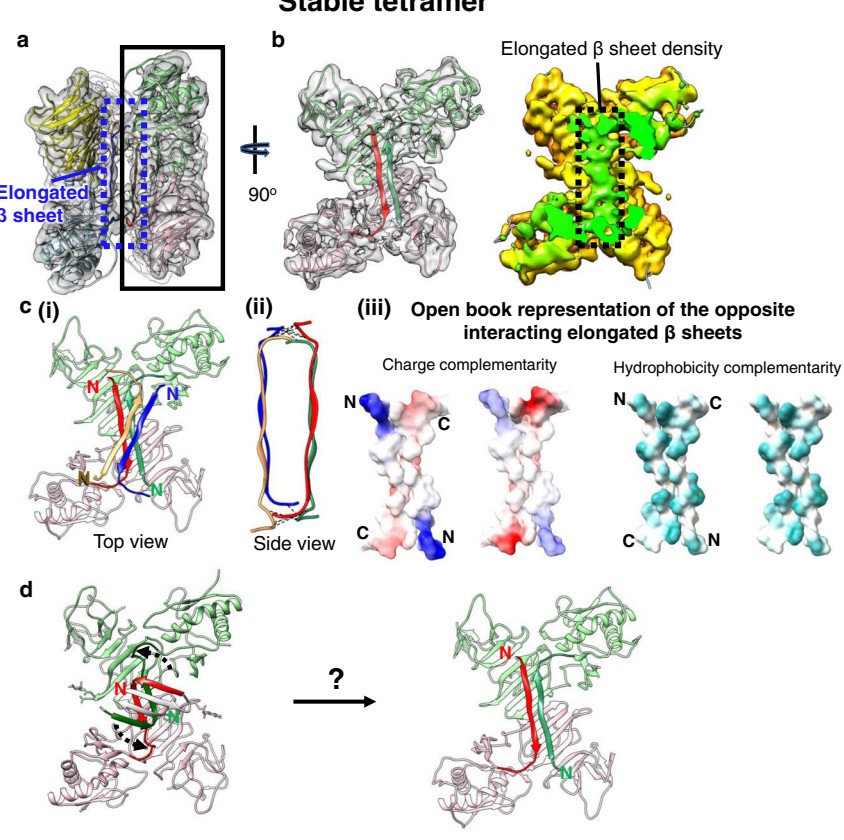

**Fig. 2 | Structure of the stable sNS1 tetramer. a** Two NS1 dimeric structures fitted into the cryoEM map (reduced sharpening was applied to make the density of the elongated β-sheet clearer). One dimer is colored in pink and light green and the other in yellow and light blue. **b** Left: the overall structure of the sNS1 dimer is similar to that of iNS1 except for the β-roll region (residues 1–30) in iNS1, which is an elongated β-sheet (residues 18–27) in sNS1. View from the elongated β-sheet of the black boxed region in **a**. The respective elongated β-strand of each protomer is highlighted in a darker shade of the same color (red and dark green). Right: surface density map with the regions corresponding to the elongated β-sheet colored in green, and the rest of the map in yellow. **c** Interaction of the elongated β-strands between the two dimers (red/green and yellow/blue). (i) Top view showing the elongated β-sheets of the red/green dimer interacting with that of the yellow/blue dimer (only its elongated β-sheet is shown). (ii) Side view showing possible hydrogen bonds or electrostatic interactions between the two elongated β-sheets from opposite dimers (dotted black lines). These interactions are identified by the distance between Cα backbone (<8 Å) and also their charge characteristics. (iii) The interacting residues are complementary in both charges and hydrophobicity. Open book representation of (ii): (Left) charge, and (Right) hydrophobicity distributions on the surface of the two elongated β-sheets. For charge, positive charges are colored in blue while negative charges are red. For hydrophobicity, cyan color indicates hydrophobic residues. **d** Comparison of iNS1 β-roll with sNS1 stable tetramer elongated β sheet and how it might change from the β-roll structure (dotted arrow on the iNS1 structure) into the elongated β sheet structure.

The overall structure of the dimer and its β-roll (Fig. 3a (ii)) is very similar to that of iNS1 (RMSD 0.8 Å), unlike the stable sNS1 tetramer. The overall 8 Å resolution loose sNS1 tetramer structure shows that the two dimers are rotated by ~35° with respect to each other and extra density is visible between the β-rolls of these two dimers (Fig. 3a (i)).

The difference between the stable and the loose tetramers is whether the residues near the N-terminus form a β-roll or an elongated β-sheet. Hence, this could be the determinant that defines stability.

An existing crystal structure of a ZIKV sNS1[12] (PDB code: 5GS6), obtained by purifying sNS1 produced using a baculovirus expression system, is a dimeric structure. However, upon examination of the crystal packing in this structure, sNS1 appears to have a tetrameric structure similar to our loose tetramer (RMSD 4.2 Å) (Fig. 3a (iii)). This may suggest that at least one of the ZIKV sNS1 oligomers might be the loose tetramer structure.

### CryoEM structure of the recombinant sNS1 hexamer
We obtained an 8 Å resolution sNS1 hexamer map (Fig. 3b (i) and Supplementary Figs. 6a, b). The low-resolution structure suggests that the NS1 molecules in the hexameric state are interacting loosely with each other. Fitting of iNS1 (Supplementary Fig. 6c) suggests that three dimers interact with each other via their wing domain and the distal end of the β-ladder. This interaction is similar to that in the loose tetramer. The density in the central core of the hexamer is weak (Supplementary Fig. 6b) and we were therefore unable to determine if the dimers have a β-roll or elongated β-sheet conformation. It is also possible that the interior of this hexamer contains lipid cargo, as suggested by other research groups[14,15] and this may prevent determination of the structure of this region.

The crystal structure of dengue dimeric iNS1[11] shows that the crystal packing contains a trimer of dimers (i.e., hexamer) organization; however, there are very few interactions between the three dimers (Fig. 3b (ii)). Their positions in the crystal lattice are mainly supported by their interactions with other iNS1 molecules inside the crystal unit cell. This suggests that the hexameric packing of iNS1 may not be physiological. Comparison of our cryoEM sNS1 hexamer to this crystal hexamer packing shows a very different organization of dimers —they are rotated by ~90° relative to each other (Fig. 3b (i) and (ii), right).

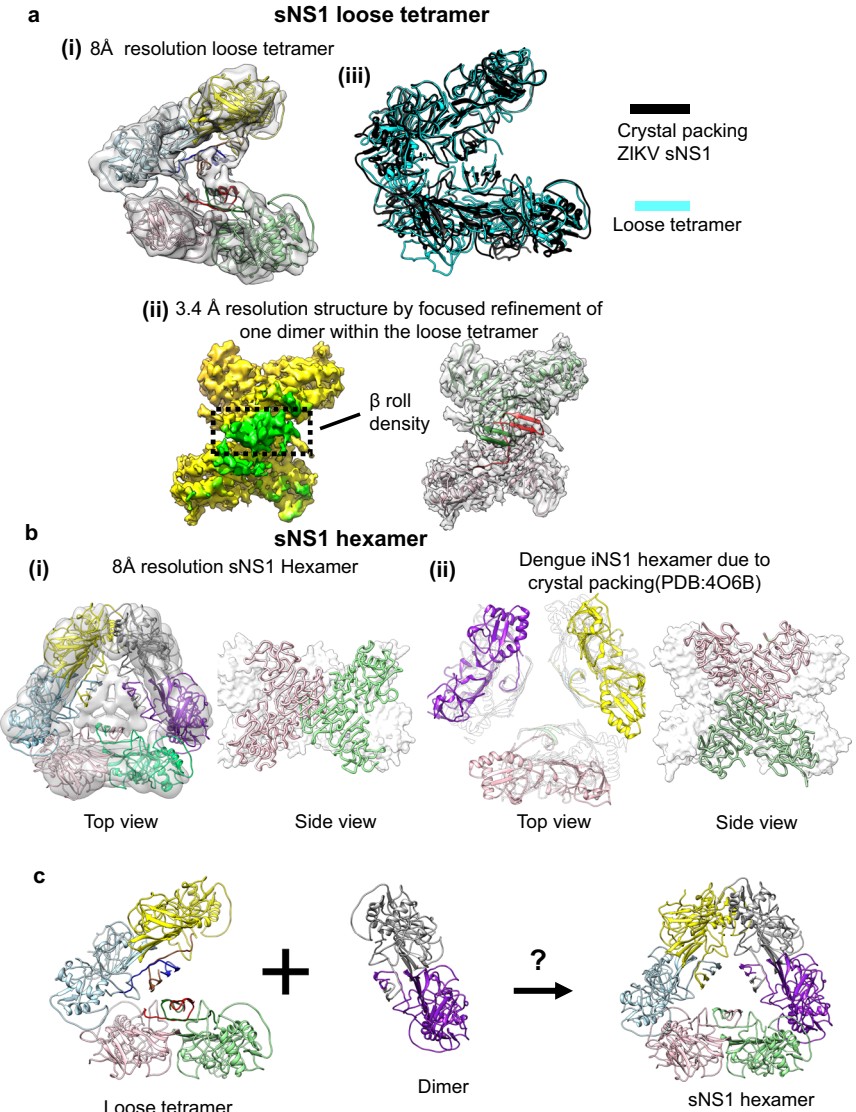

**Fig. 3 | CryoEM structures of the recombinant sNS1 loose tetramer and hexamer. a** (i) The fit of sNS1 into the 8 Å resolution loose tetramer density map. (ii) Focused refinement of one dimer of the loose tetramer yields a 3.4 Å resolution map. An iNS1 dimer fits well into the density showing that the loose tetramer contains a β-roll structure similar to that of iNS1. Left: cryoEM map with the density corresponding to the β-roll colored in green and the rest in yellow. Right: the fit of the red/green dimer into the density. (iii) Superposition of our loose tetramer with the tetramer observed in the crystal packing of ZIKV sNS1 structure (PDB: 5GS6) shows that they have similar structures. **b** Comparison of our sNS1 hexamer with the dengue iNS1 hexamer from the crystal packing. (i) Top and side views of the 8 Å resolution sNS1 hexamer structure. Left: fit of three dimers into the cryoEM map. We are unable to discern whether sNS1 adopts a β-roll or an elongated β-sheet structure, as the interior of the hexamer contains unfeatured density. (ii) The previously published dengue iNS1 structure[11] shows hexamers due to crystal packing; however, there is little interaction between the iNS1 dimers (left; top view). The dimers in the hexamer (right; side view) are also oriented differently to our hexameric structure (in (i), side view). For the side view, the protomers within one dimer are colored in pink and green ribbons, and the other dimers are shown as white surface representations. **c** Interaction of a loose tetramer with a dimer could lead to the formation of a hexamer.

It is possible that the loose tetrameric structure is a transition intermediate of the hexameric structure (Fig. 3c). However, we observed loose tetramers forming ~74% of the particle population (Fig. 1d), while hexamers only form ~3%, suggesting that the loose tetramer may be a preferred oligomerization state.

Previous work from other research groups[14,15] has shown that there are lipids associated with sNS1 that may help stabilize the overall structure. However, we observed only sparse densities near the β-rolls of the loose tetramer and the core of hexamer and are unable to ascertain that they belong to lipids. However, it is still possible that both the loose tetrameric and the hexameric structures are stabilized by lipids. To test this, we incubated the NS1 sample with detergent (0.05% DDM) and imaged it by cryoEM single particle

analysis. We observed that the percentage of loose tetramers and hexamers decreased dramatically after detergent treatment (Supplementary Table 2a), compared to untreated NS1 (Fig. 1e). This suggests that the lipids inside the loose tetramers and hexamers could be important for their structural integrity. A caveat to this conclusion is the detergent could also disrupt the hydrophobic interactions between the dimers.

**CryoEM structure of the recombinant sNS1 complexed with Fab 5E3**

We also investigated the cryoEM structure of the recombinant sNS1 complexed with the Fab fragment of an anti-NS1 antibody 5E3. The 2D class averages (Fig. 4a) of the boxed particles from the cryoEM

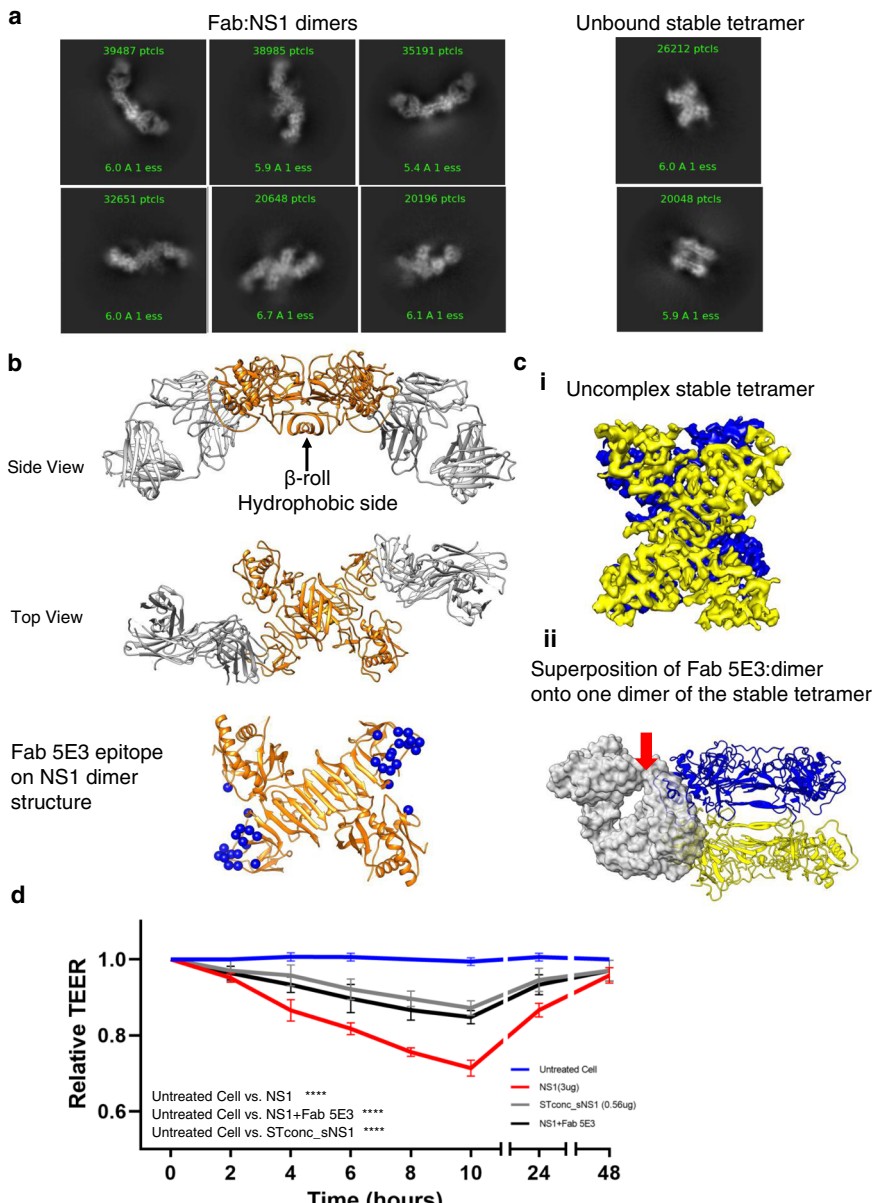

**Fig. 4 | CryoEM structure of sNS1 complexed with Fab 5E3. a** Some of the 2D class averages showing Fab bound dimers and also unbound stable tetramers. **b** The 3.5 Å resolution structure of dimer NS1(orange) complexed with Fab 5E3 (gray) shows that they form an arch shape—the Fabs are bending towards the β-roll side of the sNS1 dimer. This NS1 dimer contains the β-roll structure which is also present in the loose tetramer and the hexamer. The Fabs bind to the ends of the β-ladder. The 5E3 epitope is shown as blue spheres. **c** (i) The 3.9 Å resolution sNS1 unbound tetramer map. The two dimers of the tetramer are colored in blue and yellow. (ii) Superposition of Fab 5E3:dimer onto one dimer (yellow) of the stable tetramer shows that Fab (transparent gray surface) when bound will clash (red arrow) with the neighboring dimer (blue). **d** TEER assay showing Fab 5E3 can only partially prevent recombinant sNS1 from inducing human umbilical vein endothelial cell hyperpermeability. STconc_sNS1 (concentrated stable tetramer) has similar endothelial cell hyperpermeability activity as the sNS1 + Fab 5E3 sample. Data are the mean ± SEM from three independent experiments. Statistically significant differences between distinct groups (all time-points combined for each group) compared to the untreated groups were determined by a two-way ANOVA analysis using Dunnett's test for multiple comparisons. (**** represents $p < 0.0001$). Source data are available as a Source Data file.

micrographs suggest that, when bound by Fabs, most sNS1 particles become dimeric. Unbound stable tetramers were also present (Fig. 4a) and formed ~18% of the total particles. It suggests that the Fab 5E3 can readily dissociate the sNS1 loose tetrameric and hexameric structures. The stable tetrameric structure, on the other hand, is more resistant to Fab binding.

The structure of the Fab 5E3:sNS1 dimer complex was determined to an overall resolution of 3.5 Å (Fig. 4b, and Supplementary Fig. 7a), as measured by gold standard FSC curve cutoff at 0.143 (Supplementary Fig. 7b). The sNS1 dimer:Fab complex has an overall arch shape, where the Fabs bind to both ends of the β-ladders of the sNS1 dimer (Fig. 4b

and Supplementary Fig. 7a) and are tilted towards the β-roll side of the sNS1.

Fab 5E3 interacts with the end of the β-ladder of NS1 (Fig. 4b, Supplementary Figs. 7–8) via mainly the Fab heavy chain (Supplementary Table 3). The 5E3 epitope is similar to that in two other previously published Fab:NS1 dimer complexes[20,21]—Fabs 2B7 and 1G5.3 (Supplementary Fig. 8). Comparison of these structures shows that these three Fabs bind at different angles to the NS1 β-ladder. It has been postulated previously[11] that the hydrophobic β-roll may bind directly to cellular membranes (Supplementary Fig. 1), which could be important for its pathogenesis, and therefore the binding orientation

of Fab 5E3 may hinder this interaction. A similar neutralization mechanism was suggested for Fab 2B7[21].

We observed that these NS1 dimers have β-roll structures (Fig. 4b and Supplementary Fig. 7c). This suggests that they are likely not derived from the stable tetramer which has an elongated β-sheet structure. Hence, they could be a broken down product of Fab binding to sNS1 loose tetramers (which contain β-roll structures) and perhaps also the hexamers. We superimposed the Fab:dimer complexes into the loose tetramer and hexamer, and they showed Fabs when bound, could clash with the neighboring NS1 dimer (Supplementary Fig. 9a, b). Analysis of the solvent accessibility of the epitopes on the loose tetramer (Supplementary Fig. 9c) shows two of the epitopes (orange spheres) are partially exposed (37.5%), while the other two (cyan spheres), are fully exposed. All epitopes in the hexamer (Supplementary Fig. 9d) are partially exposed (37.5%). Since Fab likely could bind to the loose tetramer and hexamer, this suggests that the loose tetramers and hexamers likely undergo motions that will, at some time point, have their epitopes fully exposed for Fab binding, and once the Fab has bound, the loose tetramer and hexamers could be dismantled into dimers.

We reconstructed the unbound stable tetramer to 3.9 Å resolution (Fig. 4c (i)), and found that it is identical to the uncomplexed stable tetramer (Fig. 2). Analysis of the epitopes on all NS1 molecules in the unbound stable tetramer shows they are only partially exposed (37.5%) (Supplementary Fig. 9e), thus discouraging Fab binding. Superposition of the Fab 5E3:dimer complex with one of the dimers in the stable tetramer structure (Fig. 4c (ii)) also shows that the Fab, if bound, will clash with the wing domain of the neighboring dimer. While the Fab is able to cause dissociation between dimers in the loose tetramer and hexamer, it is unable to do the same to the stable tetramer. This suggests that the two dimers in the stable tetramer have tighter interactions, thus disallowing antibody binding. The presence of unbound stable tetramer is consistent with the results of the TEER assay (Fig. 4d), showing Fab can only partially inhibit sNS1-induced endothelial cell hyperpermeability. To further confirm that the stable tetramer is functional, we tried to purify the stable tetramer by harvesting the flow-through after the recombinant sNS1 sample has been exposed to the antibody 5E3:protein G magnetic beads. This sample is hereafter named, STconc_sNS1. The concentration of STconc_sNS1 sample is less than 20% of that of the original sNS1 protein. CryoEM 3D classification of the particles shows there are 75% stable tetramer and 25% loose tetramer (Supplementary Table 2b). This method successfully concentrated the stable tetramer, although some loose tetramers are present. The STconc_sNS1 sample also has similar endothelial hyperpermeability activity as when the sNS1 is complexed with Fab5E3 (Fig. 4d). This suggests the stable tetramer has hyperpermeability activity.

In conclusion, since the loose tetramer and hexamers are less stable, their quaternary structure can be easily disrupted by antibody binding. This may form part of the antibody neutralization mechanism. The stable tetramer is however more resistant to Fab 5E3 binding, and this may help sNS1 to evade the host immune response.

## Discussion

Our cryoEM and analytical ultracentrifugation studies shows the vast majority of the recombinant sNS1 particles from a DENV2 clinical strain (PVP94/07) exist as tetramers, while the hexamer only forms a minor population (Fig. 1e and Supplementary Fig. 2). Analysis of the buried surface area between two neighboring dimers within the loose and stable tetramers and that in hexamers (Supplementary Fig. 10), would indirectly suggest the affinity between the two dimers in these different oligomerization states. The two neighboring dimers in the stable and loose tetramers have a total buried area of ~650–690 Å², compared to that in the hexamer 307 Å². This could explain why there are more tetramers than hexamers.

sNS1 has been previously thought to be homogeneously hexameric in structure[14,15,22]. These studies were done by using DENV1 FGA/89 strain[15] and DENV2 sNS1 16681 strain[23] and they might have different oligomerization distributions compared to our sNS1 from DENV2 PVP94/07 clinical isolate. Nonetheless, we compared our cryoEM 2D class averages of the picked particles to that in their samples (Supplementary Fig. 11). We observed that their 2D class averages clearly showed some with top views of both loose and stable tetramers, however we cannot comment on the distribution of particles in different oligomeric states, as the number of particles in each class averages are not indicated (Supplementary Fig. 11). We also conducted a test where we carry out cryoEM image reconstruction of our particles by assuming that they are homogenously hexameric (D3 symmetry applied without further classification). The cryoEM map is similar to that reported by Gutsche et al.; however, resolution of the cryoEM map did not improve beyond 15 Å (Supplementary Fig. 12). This suggests that, at least in our sample, the particles are not homogenously hexameric.

The question on which oligomerization states (stable or loose tetrameric or hexameric) is/are the pathogenic ones is a very difficult question to answer. This is because sNS1 has been recently found to cause pathogenic effects[24] via several pathways and their mechanism is still largely unclear. Its pathogenic effect leading to vascular leakage might be due to a combination of activation of innate immunity and the direct binding effect of sNS1 to endothelial cells. Activation of innate immunity comprises complement pathway activation, and binding to toll-like receptors etc. that might eventually lead to endothelial injury. sNS1 is also thought to bind directly to endothelial cells leading to increased expression and/or activation of cathepsin L, heparinase and the endothelial sialidase which cause the disruption of the endothelial glycocalyx layer. Therefore how sNS1 and which oligomerization states cause the pathogenic effect can be complex.

Our TEER result shows that Fab 5E3 only partially inhibits hyperpermeability effect of the recombinant sNS1 on the human Umbilical Veil Endothelial cells. The cryoEM results show that Fab 5E3 binds to both loose tetramer and hexamer and dissociates them into dimers, and it does not bind to the stable tetramer. The stable tetramer has endothelial hyperpermeability activity, as demonstrated by the STconc_sNS1 sample. For the loose tetramer and hexamer, since Fab 5E3 binds to both of them and dissociates them into dimers, and the overall sNS1 endothelial hyperpermeability is lowered suggesting that either one or both of these oligomerization states are functional.

Previous analysis[25] comparing between two DENV2 strains that causes different disease severity, points to a mutation on NS1 (T164S) that leads to increased disease severity. This mutation also results in an increased in sNS1 production in mammalian cells. Analysis of this residue in our stable tetramer structure (Supplementary Fig. 13) shows it is at the interacting interface to the hydrophilic N-terminal part of the elongated β-strand of the opposite dimer. This mutation may slightly increase the hydrophilicity of this interacting interface, thus encouraging interactions between two dimers. However, this will require further investigation. Examination of this mutation on the other oligomerization states (loose tetramer and hexamer), shows it is located in the "greasy finger" adjacent to the β-roll. Since this region influence NS1 dimers interactions to each other or to lipid cargo, they might also affect the assembly of these loose tetramers and hexamers. Another mutation that can also increase the secretion of sNS1 is located that the N-terminal end (residue 11) of either the elongated β-strand or β-roll[26]. However, mutations at the N-terminal domains are rare. Sequence analysis and comparison of the flavivirus NS1 shows that the N-terminal domain (β-roll or elongated β-sheet) (residue 1–30) is highly conserved (Supplementary Fig. 4e), suggesting conservation of this region might be important for essential functions.

For cryoEM studies, we take a reductionist approach to understand the structures of sNS1 when it is expressed alone. However, in an

infected cell or human system, it is possible that once sNS1 is secreted outside the cell, it could bind with different viral/host proteins spontaneously. This might increase the number of structural classes making cryoEM structural determination much more difficult and complicated. There are examples where sNS1 bind to host proteins: Benfrid et al.[23] and Chew et al.[27] showed that sNS1 binds to human/bovine high-density lipoproteins (HDL). Benfrid et al.[23] also showed that sNS1 it binds to low-density lipoproteins, albeit with lower affinity. Benfrid et al.[23] reported negative-stain EM 2D class averages of the purified sNS1:HDL complex containing heterogeneous particles with either two, three, and four NS1 dimers on the HDL surface. They suggested how a hexameric sNS1 could rearrange into three dimers on the HDL surface. Our tetrameric structure, on the other hand, might account for the other NS1-HDL particles with either two or four dimers on the HDL surface −they might originate from one or two tetramers, respectively. Chew et al.[27], on the other hand, suggested one dimer on the surface of HDL particle, however in their work, the sNS1 was complexed with an antibody which might have disrupted the sNS1 quaternary structure, as observed from their antibody:sNS1 structure showing two Fabs binding at both ends of an NS1 dimer. This is similar to our structure of 5E3 complexed with NS1 dimeric obtained after Fab 5E3 has dismantled the loose tetramer and hexameric sNS1 structures. Both Benfrid et al.[23] and Chew et al.[27] did not show any high-resolution structures of sNS1 complexed with HDL, likely because their samples might be very heterogeneous and/or flexible, thus complicating cryoEM structural determination. Thus our cryoEM study shows the sNS1 structures before it is complexed with any host proteins.

NS1 plays multiple roles in dengue virus infection of mammalian cells and also mosquitoes. It was shown that sNS1 is one of the major factors contributing to the development of DHF/DSS in patients[5]. Our cryoEM studies show structural details of the recombinantly expressed sNS1 protein−(1) the quaternary organization of the different oligomerization states, (2) how an antibody could bind and dismantle the loose tetrameric and hexameric sNS1 structures and also (3) how sNS1 stable tetramers could resist this antibody-induced disassembly. These findings will contribute to the design of therapeutics and vaccines against severe dengue disease.

## Methods

### Cloning, expression, and purification of recombination sNS1
DENV2 PVP94/07 NS1-HisTag (a His$_6$-tag at the C-terminus) was cloned into the pcDNA 3.4-TOPO (Invitrogen) and the plasmid was transfected into Expi293F cells (Thermo Fisher). After expression in Expi293F cells, we harvested the supernatant containing sNS1. It was then clarified by centrifuging at $9000 \times g$ at 4 °C for 30 min. The sNS1 protein was purified by using a HisTrap column (GE Healthcare). It was then further purified through a gel-filtration chromatography (Superdex 200 10/300 GL column (GE Healthcare)) in buffer (50 mM Tris 8.0 and 150 mM NaCl). Protein purity was then assessed by SDS-PAGE.

For the stable tetramer sample, 150 uL of recombinant NS1 (0.38 mg/mL) was incubated with 100 μg of the antibody 5E3 bound to SureBeads Protein G Magnetic Beads (Bio-Rad) at room temperature for 3 h. The supernatant was collected for cryoEM study after magnetizing the beads.

### Purification of sNS1 from dengue virus infected cell for western Blot analysis
C6/36 cells were grown to ~80% confluency before infecting with DENV2 PVP94/07 at a multiplicity of infection of 0.5. Infected cells were incubated at 29 °C for 3 days. Tissue culture supernatant was clarified by centrifugation at $8000 \times g$ for 30 min. Virus were removed by precipitating with 8% polyethylene glycol 8000 (PEG8000) and then centrifugation to spin down the virus particles. To precipitate sNS1 from the remaining supernatant, more PEG8000 was added to achieve a final concentration of 12% and the mixture was kept for

overnight. After centrifugation, the pellet containing sNS1 was resuspended in phosphate buffer saline (PBS). To remove PEG8000 from the final sample, 200 μL of chloroform was added to 200 uL of the sNS1 sample and mixed for 10 min. It was then centrifuged for 5 min at $10,000 \times g$ to allow separation of the chloroform layer from the mixture. The aqueous phase (upper layer) containing sNS1 was collected for western blot analysis[28].

### BS³ crosslinking of protein
BS³, a crosslinker with an 11.4 Å spacer arm, were added to sNS1 (the recombinant sNS1 or the native sNS1) in PBS to a final concentration of 5 mM and the samples were incubated for 10, 20, and 30 min at room temperature. Reactions were stopped by adding Tris·HCl, pH 7.5 to a final concentration of 50 mM and then the sample was added the SDS-PAGE loading buffer, boiled for 5 min and then analyzed on a 4–20% SDS-PAGE gradient gels (Bio-Rad). The protein bands were then transferred to a nitrocellulose membrane, probed with anti-polyHistidine–peroxidase antibody (for recombinant sNS1) or anti-NS1 antibody (for sNS1 from infected cell) and visualized by chemiluminescence.

### The sedimentation velocity experiments
The sedimentation velocity experiments using analytical ultracentrifugation (SV-AUC) were performed at $72,446 \times g$ in a Beckman ProteomeLab XL-1 analytical ultracentrifuge (Beckman Coulter, 253 Brea, California, U.S.) equipped with an-60 Ti rotor at 20 °C in the vacuum. The A280 nm scan data was acquired at 6 min intervals for each sample reading. Protein solutions of 1 mg/ml were used in NTE buffer. The proteins' MW and c(s) distribution were calculated using the SEDFIT software (National Institutes of Health, Bethesda, Maryland, U.S.) to obtain the size distribution profile. The model used was "Continuous c(s) Distribution".

### CryoEM sample preparation
Quantifoil Au R1.2/1.3 grids were glow-discharged at 5 mA for 1 min using an Emitech K100X Glow Discharge Unit. About 2.1 μl of sNS1 (0.25 mg/mL) / sNS1 complexed with Fab 5E3 (molar ratio 1:1.2, final concentration of sNS1 is 0.25 mg/mL) were applied to the grids, blotted with a filter paper (Ted Pella Standard Vitrobot filter paper, Grade 595) for 2 s (Blot force 2, Blot Time 2 s, drain time 0.5 s, Humidity 100%) to remove excess sample, and then flash frozen in liquid ethane by using the Vitrobot Mark IV plunger (FEI, Netherlands). To ensure temperature consistency, the temperature of the Vitrobot humidity chamber was adjusted to 4 °C for blotting.

### CryoEM image acquisition and reconstruction procedure
CryoEM micrographs of sNS1/ sNS1 complex were collected using a Titan Krios transmission electron microscope operated at 300 kV and equipped with a Gatan K3 direct electron detector and Gatan GIF post-column energy filter operated in zero-loss mode with a slit width of 20 eV. Images were recorded at a nominal magnification of 130,000.

SerialEM was used to carry out the automated data collection. Image shift was used to target 9 exposures per stage position with an image shift of 1.6 um. Images were recorded by movie mode, with similar total exposure of 4.9 s, 58 frames per movie and total dose of 81.5 e⁻/Å². The frames from each movie were aligned using MotionCor2[29] to produce full dose images used for particle selection and orientation search. The images were taken at underfocus in the 0.8–2.0 μm range. In total, 7714 and 7145 micrographs were collected for the sNS1 and the NS1:Fab 5E3 complex, respectively. The astigmatic CTF parameters were estimated with Patch CTF estimation in CryoSPARC[17] and were accounted for during orientation search. We performed automated particle picking using Warp[30] and CryoSPARC. A total of 2,342,754 particles were selected. These particles were binned 4x before going through 20 iterations of 2D classification (CryoSPARC)

to produce 2D class averages. The starting model was generated by the CryoSPARC Ab-Initio Reconstruction. We then performed 25 iterations of 3D classification and reconstruction using C1 symmetry using the program Relion[16] to remove broken and distorted particles. We then switched to CryoSPARC to perform further 3D classification. For the class with stable tetramers, we performed multiple cycles of heterogenous and non-uniform refinement[17] by imposing D2 symmetry. For the loose tetramers, we performed heterogenous, non-uniform refinement without any symmetry imposed. We then performed local refinement on one of its dimers. For the hexamers, we performed only non-uniform refinement with D3 symmetry imposed. The cryoEM reconstruction flow chart is summarized in Supplementary Fig. 3. Resolution was determined by using gold standard Fourier shell correlation and local map resolution was estimated with ResMap[31]. The refinement statistics are shown in Supplementary Table 1.

### Protein structure building
Building of the high-resolution structures of sNS1—a dimer of the loose tetramer (PDB: 7WUS) and the stable tetramer (PDB: 7WUT) structures, were done in this sequence—(1) approximate fitting using the Zika NS1 dimer structure (PDB 5GS6) as rigid bodies into the density using "fit-in-map" function in Chimera[19], (2) mutation of residues to that of dengue NS1 sequence using the program Coot, and (3) then fitting the individual residues into densities[32]. We then refined the structures using the "phenix.real_space_refine" procedure in the Phenix software package, with default parameters and rigid body refinement using secondary-structure and torsion angle restraints[33]. The final coordinates containing an asymmetric unit (i.e., a protomer in the stable tetramer structure) were then used to build the biological unit (full tetramer) by using the command "sym" in UCSF Chimera[19]. The final coordinates of the asymmetric units were checked using MolProbity[34]. To interpret the 8 Å resolution loose tetramer (PDB:7WUU) and hexamer (PDB:7WUV) maps, we fitted these maps by using the high-resolution structure of the dimer from the loose tetramer (PDB: 7WUS). We then refined the structures using the "phenix.real_space_refine" procedure in the Phenix software package, with default parameters and rigid body refinement using secondary-structure and torsion angle restraints. Maps and structures shown in the figures were generated using UCSF Chimera and Coot.

### TEER assay
A monolayer of Human Umbilical Vein endothelial cells (80–90% confluency) cultured in T75 flasks were detached using trypsin-EDTA (0.25%). The cells were then resuspended using fresh culture media and then counted using an automated cell counter. Sixty thousand cells (300 µL per well) were seeded onto the apical side of Transwell inserts. Each Transwell was transferred inside a 24-well format plate containing 1.5 mL of endothelial cell culture media. Transwells containing endothelial cells were incubated at 37 °C and 5% $CO_2$ for 3–4 days, and 50% of the culture medium was changed in each well 48 hours post-seeding. Cells were grown until the TEER values reached ~140–160 Ohms (Ω), indicating 100% cell confluency. After this, 3 µg NS1 protein / NS1_Fab5E3 complex (molar ratio 1:1.2)/STconc_sNS1 was added to the apical side of the Transwell insert. Electrical resistance values, measured in Ohms (Ω), were recorded at every 2 h time-points following the addition of the proteins using an Epithelial Volt Ohm Meter with the "chopstick" electrodes. Endothelial permeability was expressed as relative TEER which represents a ratio of resistance values (Ω) as follows: (Ω experimental condition − Ω medium alone)/(Ω non-treated endothelial cells − Ω medium alone).

### Statistics and reproducibility
Statistical tests were performed using GraphPad Prism version 9.2.0. Variance was estimated by calculating the standard error of measurements (SEMs) and represented by error bars. All experiments were repeated at least twice, and similar results were obtained. Source data are provided as a Source Data file.

### Reporting summary
Further information on research design is available in the Nature Research Reporting Summary linked to this article.

## Data availability
The data that support this study are available from the corresponding authors upon reasonable request. The cryoEM density maps have been deposited in the Electron Microscopy Data Bank (EMDB) under accession codes EMDB-32841 (stable tetramer), EMDB-32842 (loose tetramer), EMDB-32840 (dimer of loose tetramer), EMDB-32843 (hexamer), and EMDB-32839 (sNS1:Fab5E3). The coordinates have been in the RCSB Protein Data Bank (PDB) under accession code 7WUT (stable tetramer), 7WUU (loose tetramer), 7WUS (dimer of loose tetramer), 7WUV (hexamer), and 7WUR (sNS1:Fab5E3). The previously published NS1 structures used in this study are available in the Protein Data Bank under accession codes 5GS6 and 4O6B. Data underlying Figs. 1a–c, 4d are available as a Source Data file. Source data are provided with this paper.

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

## Acknowledgements

We thank Xin-Xiang Lim, Valerie S-Y Chew, Xinni Lim, and Qunfei Zhou for help in experiments, Protein and Proteomics Centre (PPC), NUS for AUC experiment. This work was supported by Duke-NUS Signature Research Programme funded by the Ministry of Health, Singapore and National Research Foundation Investigatorship award (NRF-NRFI2016-01) awarded to S.M.L. A.W.K.T. is supported by a Khoo Postdoctoral Fellowship Award (Duke-NUS-KPFA/2021/0044) from Duke-NUS Medical School and the Estate of Tan Sri Khoo Teck Puat.

## Author contributions

S.M.L. supervised the project. S.M.L. and B.S. designed research studies; J.S.G.O. produced sNS1. W.D., J.M., and G.R.S. produced Fab. B.S. and J.S.G.O. conducted experiments. T.S.N., A.W.K.T., and J.S. collected cryoEM data; S.M.L., B.S., G.F., and V.A.K. analyzed data; S.M.L. and B.S. wrote the manuscript.

## Competing interests

The authors declare no competing interests
