## [Peer Review File · Nature Communications]

CryoEM structures of the multimeric secreted NS1, a major factor for dengue hemorrhagic feverReviewers' Comments:

Reviewer #1:

Remarks to the Author:

This manuscript presents the first high-resolution structures of secreted DENV NS1 (sNS1). Surprisingly, cryo-EM reconstructions of sNS1 reveal three distinct forms. A stable tetrameric form is predominant in the purified recombinant protein preparation, which contrasts with previous reports of hexameric forms for the native NS1. The N-terminus of this tetrameric form adopts a different fold to the one observed in the crystal structures previously determined and the loose tetrameric form determined in this study. The authors further show that the loose forms of NS1 (tetrameric or hexameric) are dissociated by Fab 5E3 but not the stable tetramer.

NS1 is highly relevant to the development of antivirals and vaccines that target severe symptoms and may not be sensitive to antibody-dependent enhancement of the disease. It remains intriguing in its function and structure, with the notable absence of a high-resolution structure of the secreted form. This study contributes exciting new structures that will be of interest to a broad readership.

The hexameric form of native NS1 is well established by different groups and using different biophysical methods (cryo-EM, SAXS, crosslinking, AUC). Very strong evidence needs to be provided to overturn this view. My concern here is that the protein has been over-expressed recombinantly with an N-terminal His-tag and without any other viral factors. It is essential that the authors rule out that one of these factors induces the formation of non-native conformations/oligomers. Furthermore, the functional and therapeutic implications of dissociation by the Fab are unclear until it has been established which form is the pathogenic one out of the dimeric, tetrameric and hexameric forms.

Major comments:

1. The protein has an N-terminal His-tag. The authors must rule out (1) that the His-tag could induce non-native oligomeric states/conformations and (2) that the purification method could bias the relative ratios observed (e.g. failure to purify hexameric NS1 because the His-tag is buried). It is notable that the other study cited to have primarily tetramers (l. 349-350) is based on a modified NS1 fused to GFP, a bulky addition at the N-terminus end.
2. The protein is over-expressed recombinantly in 293 cells. It is likely that the kinetics and levels of NS1 expression differ markedly from expression in the context of a viral infection. In turn, this is likely to result in non-native events (aberrant oligomeric states, altered glycosylation patterns). Unless excluded experimentally, this point should be discussed in all relevant places (abstract, introduction and discussion).
3. A couple of points are incorrect:
 - As performed here, chemical crosslinking cannot be used to determine quantitatively the ratio of oligomeric forms. It is possible that chemical crosslinking is weak for the hexamer due to the lack of crosslinkable residues within an appropriate distance, not necessarily because of lower levels of the hexamer.
 - When referring to previously published SAXS data, the authors compare the radius of gyration (R_g) with the maximum dimension of the tetramer/hexamer (D_{max}). These are distinct features of a molecule. SAXS has the power to indicate monodispersity and distinguish a tetramer from an hexamer. SAXS curves can be calculated from the existing structure providing a theoretical R_g for the hexamer and tetramers, and a target to fit with experimental data (should it be deposited in public databases or generated by the authors).
4. The functional assay seems to assume that the tetrameric/hexameric forms are the pathogenic ones. It is entirely possible that
 - (1) the stable tetrameric form is a non-functional form of the protein (or one that's only functional in

one of the two known functions of NS1, tissue permeability and cytokine induction).

(2) the functional form is a dimer generated from the other forms. In this case, antibodies that simply disrupt the tetramer/hexamer will have no intrinsic functional effect unless they block another functional interaction (e.g. NS1/receptor).

The result section needs to take this into account and the conclusions on therapeutics/vaccines toned down to reflect unknowns around the active form of NS1 in pathogenesis.

Minor comments:

l. 4, "multi-oligomeric": unclear

l. 39, "correlates closely with the severity of the disease": is this really that clear? Additional references are needed to support this statement.

l. 165, "long anti-parallel beta-sheet": how long? Please provide a Zoom of this density in Fig. 2 so that its connectivity and quality can be clear to the reader (similar to ED Fig4D but for the extended beta-sheet).

l. 170-174: electrostatic and hydrophobic surface analyses are risky given the low resolution in this area. Just advise caution in interpretation as side chains may be missing or mismodelled.

l.320 (or earlier): it would be useful to have a comparison of buried surfaces. While, energetics of the various assemblies are likely to be incorrect due to the low resolution of the loose tetramer and hexamer, the buried surface calculations should be fairly reliable using the maps and/or models.

Methods:

It looks like only the asymmetric unit was refined. For the stable tetramer, a full-refinement is possible and preferable to account for interface residues and reduce inter-molecular clashes.

The overall clash scores are high for 7WUS, 7WUR and 7WUT also reflected by several pages of reported overlaps. Poor fit is flagged for residues 108-130.

Reviewer #2:

Remarks to the Author:

In the manuscript "CryoEM structures of the multimeric secreted NS1, a major factor for dengue hemorrhagic fever", the authors characterized different conformations of a promising therapeutic target of Dengue virus --NS1 protein. It is of interest in the field and tends to fill a gap in the Flavivirus NS1 research. However, some questions need to be addressed, and I hope these concerns listed below can help the authors to improve the manuscript.

Major comments:

1. The main discovery of the study is the conformational change between the "elongated β -sheet" and " β -roll" of NS1 N-termini. Therefore, it is very important to illustrate the elongated β -sheet and β -roll (models) with their local EM densities (maps), which can demonstrate the reliability of the models and the new theory. Across all the figures, I can see the β -roll fits well with the EM densities (Extended Data Fig. 5B and Extended Data Fig. 7C), but I didn't find the same view for the elongated β -sheet (Fig.2B is zoomed-out and unclear).

2. Many density maps illustrated in the figures of the manuscript were fitted by models, however, in the legends, the authors didn't indicate which model(s) were used for map fitting (e.g., Fig. 2 and Fig.3), the new models solved in the manuscript or previously published models? Besides, in the Extended Data Fig. 1A legend, it was written as: "(A) Crystal structure of dengue dimeric iNS1 (PDB:5GS6)." But the PDB: 5GS6 is actually a Zika virus NS1 model. Therefore, the authors should thoroughly clarify the models used in the manuscript, which is critical for a structures-based paper.

3. Line 457 (Data availability): several EM-maps of this study are reconstructed with low-resolutions ($\sim 8\text{\AA}$), such as the Loose tetramer (EMD: 32840 and PDB: 7WUS), hexamer (EMD: 32842 and PDB: 7WUV), and authors also built models for the low-res maps. It's quite confusing when I am reading, and I would say no one can build models with such maps. Did the author really build models with the maps, or just fitted models in? If a model is not built with a map, it should have no PDB code. A more rigorous solution is: to consider just uploading the low-res maps without models.

4. A previous publication (Sci Transl Med 2019; PMID: 31243154) demonstrated the Dengue NS1 mutation T164S, which is located at the interface of NS1 tetramer, alters the disease severity in mice, and the NS1 hexamer was predicted to be less stable with this mutation. It is commended the authors can cite and discuss the role of this mutation with the "elongated β -sheet" discovery. It may extend the interest and improve the quality of the manuscript.

5. For the putative densities of lipids, basically, I suppose it is hard to state there are lipids with the maps, especially when they were refined with D2 or D3 symmetry.

Minor comments:

1. Please use correct and standard descriptions in virology.

Line 15: "...significant human pathogens like West Nile (WNV), yellow fever (YF) and Zika virus (ZIKV)..." The pathogens are "West Nile virus" and "Yellow Fever virus", but not "West Nile" and "yellow fever". Yellow Fever (YF) is the name of the infectious disease, not the name of the pathogen. The sentence may be written as "significant human pathogens, such as West Nile virus (WNV), Yellow Fever virus (YFV) and Zika virus (ZIKV)".

2. Line 427 (Protein structure building): just feel curious why the authors used a Zika NS1 model as the initial model, since there are several published Dengue NS1 models (e.g., 4O6B).

Reviewer #3:

Remarks to the Author:

The Dengue virus non-structural protein 1 (NS1) is a multifunctional protein. Inside infected cells, NS1 is involved in viral replication, by associating with cellular membranes to form the viral replication complexes. Additionally, NS1 can also be secreted, and the amount of secreted NS1 has been linked with Dengue hemorrhagic fever. While there are structures available of intracellular NS1 (by X-ray crystallography), there are no structures for soluble NS1.

In this manuscript Shu et al study complexes of secreted dengue virus NS1 protein by cryo-EM. They are able to solve the structure of stable and loose tetramers, and of hexameric structures of NS1 (to a lower resolution). The authors also solve the structure of secreted NS1 in the presence of a Fab, which appears to only bind to loose tetramers or to hexamers; not to stable tetramers. Comparing the structures of the previously published intracellular tetramers with the stable and loose secreted tetramers, the authors found a key difference in the N-terminal domain conformation: it adopts an elongated beta-sheet in the stable tetramer; and a beta-roll in the loose tetramers and the hexamers.

While the manuscript is clearly written and the results are solid, I fail to understand how the structure of soluble NS1 helps understand its effect on Dengue hemorrhagic fever. The authors could address this issue by performing TEER in the presence of stable tetramers compared to loose tetramers. Additionally, the Fab used by the authors only partially prevents NS1 from inducing human umbilical vein endothelial cell hyperpermeability; but according to Fig 1E only $\sim 23\%$ of the particles are in a stable tetramer conformation. So the observation that the Fab cannot bind stable tetramers does not seem to explain the low effect seen by the Fab (I would expect more protection if the Fab was blocking the effect of $\sim 77\%$ of NS1).

Minor comments:

- Figure 1B requires a loading control of some sort; a sample incubated 30 min with a crosslinker should not have more monomer than when it is incubated only 10 min.
- It is unclear what the authors are showing in the "Side view" panels of Fig 3Bii. Why is there so much unoccupied density? Or is it the other dimer represented as a surface?

REVIEWER COMMENTS

Reviewer #1 (Remarks to the Author):

This manuscript presents the first high-resolution structures of secreted DENV NS1 (sNS1). Surprisingly, cryo-EM reconstructions of sNS1 reveal three distinct forms. A stable tetrameric form is predominant in the purified recombinant protein preparation, which contrasts with previous reports of hexameric forms for the native NS1. The N-terminus of this tetrameric form adopts a different fold to the one observed in the crystal structures previously determined and the loose tetrameric form determined in this study.

The authors further show that the loose forms of NS1 (tetrameric or hexameric) are dissociated by Fab 5E3 but not the stable tetramer.

NS1 is highly relevant to the development of antivirals and vaccines that target severe symptoms and may not be sensitive to antibody-dependent enhancement of the disease. It remains intriguing in its function and structure, with the notable absence of a high-resolution structure of the secreted form. This study contributes exciting new structures that will be of interest to a broad readership.

(1)The hexameric form of native NS1 is well established by different groups and using different biophysical methods (cryo-EM, SAXS, crosslinking, AUC). Very strong evidence needs to be provided to overturn this view.

Firstly, our samples are not exactly the same: previous studies used the DENV1 FGA/89 strain (Gutsche et al., 2011)¹ and DENV2 sNS1 16681 strain (Benfrid et al., 2022)², whereas we used a DENV2 PVP94/07 clinical isolate, so the distribution of oligomerization states might differ between these strains.

Secondly, we do not think that previously reported cryo-EM structures are conclusive proof that the hexameric form of NS1 is the only form possible. The resolution of those structures was very limited (~30Å at best), leading to possible ambiguity in their interpretation as protein secondary structure is not visible in the maps at this resolution. In contrast, we have reported maps here at resolutions high enough to allow reliable fitting of both polypeptide backbone and side chain densities, making it unlikely that our reported tetramers are simply artifacts of incorrect reconstruction of hexamers. If the sample is assumed to be homogeneously hexameric when it is not, the result will be a low resolution cryo-EM reconstruction, as averaging different structural states will lead to smearing of densities. The general view in the cryoEM field is that the correctness of low resolution structures may be ambiguous without further verification, for example by comparing tilt pair images etc. This was not done in previous publications.

In order to address these concerns, we tried to obtain a reconstruction from our dataset (which contains different oligomerization states) by assuming that the sample is homogeneously hexameric (no classification was done and the map was made by imposing D3 symmetry). The final map obtained could not be improved beyond 15 Å in resolution and appears similar to that in Gutsche et al., 2011¹. This suggests that further classification of the particles in the dataset from Gutsche et al., 2011 may result in similar structures to those reported in our manuscript.

Test: our data treat as homogenous hexamer for reconstruction - no classification and D3 symmetry applied

We did not perform SAXS or AUC experiments with our sample and hence cannot comment on how our sample compares to others³. However, we can compare our dataset with the cryoEM or negative stained 2D class averages of particles from previous publications which had also been analyzed by either SAXS and AUC or AUC only (Gutsche et al., 2011³ and Benfrid et al., 2022²)— see figure below. We observed some 2D class averages showing clear top views of both stable and loose tetramers. This suggests that their samples were not homogeneously hexameric.

Comparison of some of the 2D class averages from (a) our sample with that in (c) Gutsche et al., 2011¹ and (d) Benfrid et al., 2022². (b) Projections made from our cryoEM maps of the stable and loose tetramers and also the hexamer representing some of the side and top views

of each structure. Possible similar projections are boxed in the same color. In (d), the unboxed class averages are largely the different orientations of the side views.

We have added this into our discussion (see below) and the above figures (new Supplementary Figs. 10 & 11) in the supplementary information.

“sNS1 has been previously thought to be homogeneously hexameric in structure. These studies were done by using DENV1 FGA/89 strain (Gutsche et al., 2011) and DENV2 sNS1 16681 strain (Benfrid et al., 2022) and they might have different oligomerization distributions compared to our sNS1 from DENV2 PVP94/07 clinical isolate. Nonetheless, we compared our cryoEM 2D class averages of the picked particles to that in their samples (Supplementary Fig. 10). We observed that their 2D class averages clearly showed some with top views of both loose and stable tetramers, however we cannot comment on the distribution of particles in different oligomeric states, as the number of particles in each class averages are not indicated (Supplementary Fig. 10). We also conducted a test where we carry out cryoEM image reconstruction of our particles by assuming that they are homogeneously hexameric (D3 symmetry applied without further classification). The cryoEM map is similar to that reported by Gutsche et al., 2011, however, resolution of the cryoEM map did not improve beyond 15Å (Supplementary Fig. 11). This suggests that, at least in our sample, the particles are not homogeneously hexameric.”

As for cross-linking gel experiments, our result is similar to previous published manuscripts^{1,3}, which also showed bands corresponding to hexamer, tetramer, dimer and monomers – see below.

Above figure: Cross-linking SDS-PAGE gel from other publications: (A) Muller et al., 2012⁴ who use the same cross-linker (BS³) as ours. The recombinant DENV2 sNS1 was expressed by using the HEK cell expression system, similar to our system. (B) DENV1 sNS1 crosslink experiment by Flamand et al., (1999)³ using crosslink agent: DMS. The sNS1 was isolated by pulling down and purifying sNS1 from DENV1 infected cells using antibodies. (C-D) Our crosslinking experiments with (C) recombinant DENV2 sNS1 and (D) sNS1 from DENV2 infected cell supernatant. For (D), the supernatant of DENV2 infected cell was PEG precipitated before conducting crosslinking experiments.

Results from these crosslink experiments of recombinant DENV2 from (A) Muller et al., 2012⁴ and (C) ours are largely similar showing bands corresponding to hexamer, tetramers, dimers and monomers. As for sNS1 from infected cells, (B) Flamand et al., 1999³ shows their DENV1 sNS1 has bands corresponding to dimers, tetramers and hexamers, while our DENV2 infected cell supernatant shows bands corresponding to tetramers and hexamers. Both BS³ and DMS spacer length is about 11Å.

(2) My concern here is that the protein has been over-expressed recombinantly with an N-terminal His-tag and without any other viral factors. It is essential that the authors rule out that one of these factors induces the formation of non-native conformations/oligomers.

We thank the reviewer for pointing this out. We forgot to mention that His tag was placed at the C-terminal end of NS1, not the N-terminal end. We have now mentioned this in the main text and also the Methods section.

Main text:

From

“His-tagged NS1 (DENV2 clinical strain PVP94/07) was expressed in human embryonic kidney (HEK) cells.”

To

“NS1 with His-tag at its C-terminus (DENV2 clinical strain PVP94/07) was expressed in human embryonic kidney (HEK) cells.

Methods section:

From

“DENV 2 PVP94/07 NS1-HisTag was cloned into the pcDNA 3.4-TOPO (Invitrogen) and the plasmid was transfected into Expi293 HEK cells (Thermo Fisher).”

to

“DENV 2 PVP94/07 NS1-HisTag (a His₆-tag at the C-terminus) was cloned into the pcDNA 3.4-TOPO (Invitrogen) and the plasmid was transfected into Expi293 HEK cells (Thermo Fisher).”

Analysis of the position of C-terminal ends in our structures of the oligomers (the stable tetramer, loose tetramer, and hexamer) shows they likely will not interfere with sNS1 dimer to dimer interactions. See the figure below- green stars (*) indicate where the C-terminal ends are of each NS1 protein.

As for viral factors that may contribute to folding of sNS1, the samples prepared for (1) cryoEM studies in the Gutsche et al., 2011 paper¹, used sNS1 from DENV1 infected cells, (2) for negative stained EM in the Benfrid et al., 2022 paper², they used the recombinant DENV2 sNS1 expressed in a Drosophila cell line, and (3) ours was expressed in a HEK cell line. As

discussed earlier, the 2D class averages showing different oligomeric states are very similar across all these samples, therefore we do not think viral factors would introduce significant changes to the sNS1 oligomerization states. However, there are two papers describing interactions of sNS1 with host (human or bovine) HDL (Benfrid et al., 2022² and Chew et al., 2022⁵). This likely occurs after the sNS1 is secreted outside the cell. We have included this in our discussion:

“For cryoEM studies, we take a reductionist approach to understand the structures of sNS1 when it is expressed alone. In an infected cell or human system, it is possible that once sNS1 is secreted outside the cell, it could bind with different viral/host proteins spontaneously. This might increase the number of structural classes making cryoEM structural determination much more difficult and complicated. There are examples where sNS1 bind to host proteins: Benfrid et al., 2022 and Chew et al., 2022 showed that sNS1 binds to human/bovine high-density lipoproteins (HDL). Benfrid et al., 2022 also showed that sNS1 binds to low-density lipoproteins, albeit with lower affinity. Benfrid et al., 2022 reported cryoEM 2D class averages of the purified sNS1:HDL complex containing heterogeneous particles with either two, three and four NS1 dimers on the HDL surface. They suggested how a hexameric sNS1 could rearrange into three dimers on the HDL surface. Our tetrameric structure, on the other hand, might account for the other NS1-HDL particles with either two or four dimers on the HDL surface—they might originate from one or two tetramers, respectively. Chew et al., 2022, on the other hand, suggested one dimer on the surface of HDL particle, however in their work, the sNS1 was complexed with an antibody which might have disrupted the sNS1 quaternary structure, as observed from their antibody:sNS1 structure showing two Fabs binding at both ends of an NS1 dimer. This is similar to our structure of 5E3 complexed with NS1 dimeric obtained after Fab 5E3 has dismantled the loose tetramer and hexameric sNS1 structures. Both Benfrid et al., 2022 and Chew et al., 2022 did not show any high resolution structures of sNS1 complexed with HDL, likely because their samples might be very heterogeneous and/or flexible, thus complicating cryoEM structural determination. Thus our cryoEM study shows the sNS1 structures before it is complexed with any host proteins.”

(3) Furthermore, the functional and therapeutic implications of dissociation by the Fab are unclear until it has been established which form is the pathogenic one out of the dimeric, tetrameric and hexameric forms.

Yes, we agree with the reviewer that the question on which oligomerization states (tetrameric or hexameric) is/are the pathogenic ones is a very difficult question to answer. This is because sNS1 has been only recently found to cause pathogenic effect⁶ and the mechanism is still largely unclear. Some research suggests that its pathogenic effect leading to vascular leakage is due to a combination of immune reactions and direct effect of binding to endothelial cells. Immune reactions involve activation of innate immunity (complement pathway, binding to toll-like receptors, etc) leading to endothelial cell layer injury. sNS1 is also thought to bind directly to endothelial cells leading to increased expression and/or activation of cathepsin L; heparinase and the endothelial sialidase which cause the disruption of the glycocalyx layer. Therefore how sNS1 causes its pathogenic effect can be extremely complex. It is thus not within the scope of this paper to identify which oligomeric states lead to which pathway of pathogenesis. However, our findings do open up the research in this area for further investigations.

Our TEER result shows that Fab 5E3 only partially inhibits hyperpermeability effect of sNS1 on the human Umbilical Veil Endothelial cells. Our CryoEM result shows that Fab 5E3 binds to both loose tetramers and hexamers and dissociate them into dimers, and it does not bind to the stable tetramer. The observation that some sNS1 hyperpermeability activity remains after antibody treatment suggests that the unbound stable tetramer causes this and hence the stable tetramer has hyperpermeability activity. Fab 5E3 binds to loose tetramer and hexamer and dissociate them into dimers, the hyperpermeability effect of sNS1 is partially inhibited, this suggests either one or both of these oligomerization states can cause hyperpermeability.

We have included this in the discussion section.

“The question on which oligomerization states (stable or loose tetrameric or hexameric) is/are the pathogenic ones is a very difficult question to answer. This is because sNS1 has been recently found to cause pathogenic effects via several pathways and their mechanism is still largely unclear. Its pathogenic effect leading to vascular leakage might be due to a combination of activation of innate immunity and the direct binding effect of sNS1 to endothelial cells. Activation of innate immunity comprises complement pathway activation, and binding to toll-like receptors etc, that might eventually lead to endothelial injury. sNS1 is also thought to bind directly to endothelial cells leading to increased expression and/or activation of cathepsin L, heparinase and the endothelial sialidase which cause the disruption of the endothelial glycocalyx layer. Therefore, how sNS1 and which oligomerization states cause the pathogenic effect can be extremely complex.

Our TEER result shows that Fab 5E3 only partially inhibits hyperpermeability effect of sNS1 action on the human Umbilical Veil Endothelial cells. The cryoEM results show that Fab 5E3 binds to both loose tetramer and hexamer and dissociates them into dimers, and it does not bind to the stable tetramer. The observation that some sNS1 hyperpermeability activity remains after antibody treatment, suggests that the unbound stable tetramer causes this and hence the stable tetramer has hyperpermeability activity. As Fab 5E3 binds to loose tetramers and hexamers and dissociates them into dimers, and because the hyperpermeability effect of sNS1 is partially inhibited, this suggests that either one or both of these oligomerization states can cause hyperpermeability.”

Major comments:
(4)1. The protein has an N-terminal His-tag. The authors must rule out (1) that the His-tag could induce non-native oligomeric states/conformations and (2) that the purification method could bias the relative ratios observed (e.g. failure to purify hexameric NS1 because the His-tag is buried). It is notable that the other study cited to have primarily tetramers (l. 349-350) is based on a modified NS1 fused to GFP, a bulky addition at the N-terminus end.

Thank you for pointing this out. Please see reply to the question (2). The His-tag is on the C-terminal end of the NS1, and analysis of the hexameric structure shows the His-tag should be exposed. We have now deleted the sentence on the yellow fever NS1 that is fused to GFP.

(5)2. The protein is over-expressed recombinantly in 293 cells. It is likely that the kinetics and levels of NS1 expression differ markedly from expression in the context of a viral infection. In turn, this is likely to result in non-native events (aberrant oligomeric states, altered glycosylation patterns). Unless excluded experimentally, this point should be discussed in all relevant places (abstract, introduction and discussion).

Following the advice of reviewer #1, we have inserted the word “The recombinant sNS1”, throughout the manuscript to remind readers that this is a recombinant protein and not from infected cell culture.

We thank the reviewer for asking this. This is a very interesting and challenging question. It probably applies to all crystal and cryoEM structures that were most often produced by expressing recombinant proteins as otherwise, it is difficult to get enough proteins to conduct these structural experiments. Nevertheless, they have been proven to be useful.

Our TEER assay shows that our recombinant sNS1 has endothelial cell hyperpermeability activity suggesting that it is functional.

It is hard to purify large amounts of native sNS1 from infected dengue virus culture and the harsh purification conditions might also alter the protein structure. To purify sNS1, we need to make large scale cultures and then the supernatant is purified via antibody-affinity column (Gutsche et al., 2011¹). In order to elute sNS1 from the column, a very low pH buffer (pH 2.7) is used. One could also argue that the low pH treatment might alter the sNS1 quaternary assembly. In addition, from our work, we observe antibody binding could break the loose tetramer and hexamers apart. Therefore, it is also possible that during antibody affinity column purification, the higher oligomers are broken into dimers, and then after elution and pH neutralization, they might try to re-assemble into higher oligomeric forms, the highly concentrated sNS1 proteins might also influence the reassembly process – this might not represent the physiological conditions too.

As for glycosylation patterns, since our sNS1 is expressed in human cell line, the glycosylation is likely similar to that in a natural infection in humans.

(6)3. A couple of points are incorrect:

- As performed here, chemical crosslinking cannot be used to determine quantitatively the ratio of oligomeric forms. It is possible that chemical crosslinking is weak for the hexamer due to the lack of crosslinkable residues within an appropriate distance, not necessarily because of lower levels of the hexamer.

It is possible that one could interpret the lower bands (tetramers, dimers and monomers) as a result of incomplete cross-linking. But it is also possible that there is truly a mixture of different oligomerization states and therefore, there are bands corresponding to them. However, it is probable that the result is a mixture of all these – presence of different oligomerization states and also incomplete crosslinking.

We agree with reviewer that chemical crosslinking should not be used to quantify the percentage of different oligomerization states. We have deleted the following sentence from the result section.

~~“These results also showed that the tetrameric form is the predominant oligomerization state in the NS1 protein sample.”~~

- When referring to previously published SAXS data, the authors compare the radius of gyration (Rg) with the maximum dimension of the tetramer/hexamer (Dmax). These are distinct features of a molecule. SAXS has the power to indicate monodispersity and distinguish a tetramer from an hexamer. SAXS curves can be calculated from the existing structure

providing a theoretical R_g for the hexamer and tetramers, and a target to fit with experimental data (should it be deposited in public databases or generated by the authors).

We thank the reviewer for pointing out that the R_g measurement from SAXS data is different from the maximum dimension (D_{max}) that we observed from the cryoEM structure. We have removed that from our discussion.

(7)4. The functional assay seems to assume that the tetrameric/hexameric forms are the pathogenic ones. It is entirely possible that (1) the stable tetrameric form is a non-functional form of the protein (or one that's only functional in one of the two known functions of NS1, tissue permeability and cytokine induction).

(2) the functional form is a dimer generated from the other forms. In this case, antibodies that simply disrupt the tetramer/hexamer will have no intrinsic functional effect unless they block another functional interaction (e.g. NS1/receptor).

The result section needs to take this into account and the conclusions on therapeutics/vaccines toned down to reflect unknowns around the active form of NS1 in pathogenesis.

Please see the answer to question (3), which addresses this point.

Minor

comments:

(1) l. 4, "multi-oligomeric": unclear

We have modified the abstract.

From

"Dengue virus infection can cause dengue hemorrhagic fever (DHF). Dengue NS1 is multifunctional: the intracellular dimeric NS1 (iNS1) forms part of the viral replication complex, the extracellular multi-oligomeric secreted NS1 (sNS1) is a major factor contributing to DHF. The structure of the iNS1 is well studied but not sNS1. Here we show the tetrameric (stable and loose conformation) and hexameric structures of the recombinant sNS1. Stability of the stable and loose tetramers is determined by the conformation of their N-terminal domain – elongated β -sheet or β -roll. Binding of an anti-NS1 Fab breaks the loose tetrameric and hexameric sNS1 into dimers, whereas the stable tetramer remains largely unbound. Our results show detailed quaternary organization of different oligomeric states of sNS1 and will contribute towards the design of dengue therapeutics."

To

"Dengue virus infection can cause dengue hemorrhagic fever (DHF). Dengue NS1 is multifunctional. The intracellular dimeric NS1 (iNS1) forms part of the viral replication complex. Previous studies suggest the extracellular secreted NS1 (sNS1), which is a major factor contributing to DHF, exists as hexamers. The structure of the iNS1 is well studied but not sNS1. Here we show by cryoEM that the recombinant sNS1 exists in multiple oligomeric states: the tetrameric (stable and loose conformation) and hexameric structures. Stability of the stable and loose tetramers is determined by the conformation of their N-terminal domain

– elongated β -sheet or β -roll. Binding of an anti-NS1 Fab breaks the loose tetrameric and hexameric sNS1 into dimers, whereas the stable tetramer remains largely unbound. Our results show detailed quaternary organization of different oligomeric states of sNS1 and will contribute towards the design of dengue therapeutics.”

(2) I. 39, “correlates closely with the severity of the disease”: is this really that clear? Additional references are needed to support this statement.

We thank the reviewer for pointing it out. We have checked on this.

Here is the paper showing that the secreted sNS1 levels in plasma correlated with viremia levels and were higher in patients with DHF than in those with DF.

Reference paper: High Circulating Levels of the Dengue Virus Nonstructural Protein NS1 Early in Dengue Illness Correlate with the Development of Dengue Hemorrhagic Fever, *The Journal of Infectious Diseases*, Volume 186, Issue 8, 15 October 2002, Pages 1165–1168.

But we also found other papers that contradict this:

Correlation of host inflammatory cytokines and immune-related metabolites, but not viral NS1 protein, with disease severity of dengue virus infection. *PloS one*, 15(8), p.e0237141.

Therefore, we have decided to delete this sentence “*In vivo* studies show that the sNS1 is present at very high concentrations in patient sera and correlates closely with the severity of disease.”

(3) I. 165, “long anti-parallel beta-sheet”: how long? Please provide a Zoom of this density in Fig. 2 so that its connectivity and quality can be clear to the reader (similar to ED Fig4D but for the extended b-sheet).

Thank the reviewer for this advice. The beta-sheet is 27 Å long (Residue 18-27). We have changed this in the manuscript.

“Fitting of the density map shows that residues 18 to 27 form a 27 Å long β -strand which pairs with the same β -strand of the other protomer within the dimer to form a long anti-parallel β -sheet”

As this elongated β -strand is sandwiched in the middle of the stable tetramer, it is hard to show it as a figure. We have made a supplementary video (Supplementary movie 1) to show the structure and density of long anti-parallel beta-sheet region.

(4) I. 170-174: electrostatic and hydrophobic surface analyses are risky given the low resolution in this area. Just advise caution in interpretation as side chains may be missing or mismodelled.

Thank the reviewer for this advice, we have removed all the residue labels in the figure2. We will just point that they have charge and hydrophobic complementarity, instead of indicating which residue is interacting with which.

(5) I.320 (or earlier): it would be useful to have a comparison of buried surfaces. While, energetics of the various assemblies are likely to be incorrect due to the low resolution of the loose tetramer and hexamer, the buried surface calculations should be fairly reliable using the maps and/or models.

We thank the reviewer for this advice. The calculation of the buried areas between the two neighboring dimers in the stable and loose tetramers and also the hexamers, showed that they likely have higher affinity in the tetrameric state than in the hexameric state. We have included a new supplementary figure on this (Supplementary Fig. 9) and also inserted this in the discussion.

New figure:

New discussion:

“Our cryoEM study shows the vast majority of the recombinant sNS1 particles from a DENV2 clinical strain (PVP 94/07) exist as tetramers, while the hexamer only forms a minor population (Fig. 1E). Analysis of the buried surface area between two neighboring dimers within the loose and stable tetramers and that in hexamers (Supplementary Fig. 9), would indirectly suggest the affinity between the two dimers in these different oligomerization states. The two neighboring dimers in the stable and loose tetramers have a total of ~650 to 690Å², compared to that in the hexamer 307Å². This could explain why there are more tetramers than hexamers.”

Methods:

(6) It looks like only the asymmetric unit was refined. For the stable tetramer, a full-refinement is possible and preferable to account for interface residues and reduce inter-molecular clashes.

Thank you for this advice. We have released the symmetry for the tetramer map from D2 to C1. We compared densities of the elongated β -sheet region in these two maps and they look largely the same, except with C1 symmetry, the resolution of the densities are slightly poorer – see below figure.

(7) The overall clash scores are high for 7WUS, 7WUR and 7WUT also reflected by several pages of reported overlaps. Poor fit is flagged for residues 108-130.

We thank the reviewer for pointing it out. We have now reduced the number of clashes.

For the tetramer model (7WUT), we have deleted residues 116-126 as the corresponding density in this region is very poor and therefore the fit is not accurate. We optimize the overall fit and Clashscore has improved from 32 to 14.

For the dimer of loose tetramer (7WUS) and the NS1:Fab5E3 complex (7WUR), the Clashscore drops to 11.

We have also examined these remaining clashes, they are mostly between autogenerated hydrogen atoms and the clash distances are 0.4-0.7Å.

Reviewer #2 (Remarks to the Author):

In the manuscript “CryoEM structures of the multimeric secreted NS1, a major factor for dengue hemorrhagic fever”, the authors characterized different conformations of a promising therapeutic target of Dengue virus --NS1 protein. It is of interest in the field and tends to fill a gap in the Flavivirus NS1 research. However, some questions need to be addressed, and I hope these concerns listed below can help the authors to improve the manuscript.

Major

comments:

1. The main discovery of the study is the conformational change between the “elongated β -sheet” and “ β -roll” of NS1 N-termini. Therefore, it is very important to illustrate the elongated β -sheet and β -roll (models) with their local EM densities (maps), which can demonstrate the reliability of the models and the new theory. Across all the figures, I can see the β -roll fits well with the EM densities (Extended Data Fig. 5B and Extended Data Fig. 7C), but I didn’t find the same view for the elongated β -sheet (Fig.2B is zoomed-out and unclear).

Thank you for this advice. We have added a video (Supplementary movie 1) to show this region in detail.

2. Many density maps illustrated in the figures of the manuscript were fitted by models, however, in the legends, the authors didn’t indicate which model(s) were used for map fitting (e.g., Fig. 2 and Fig.3), the new models solved in the manuscript or previously published models?

Thank the reviewer for pointing it out. The models used for map fitting in the Fig.2 and Fig.3 are the new models solved in this manuscript.

In detail, in figure 2, to build the tetramer model (PDB: 7WUT), we use full-length zika NS1(PDB: 5GS6) to fit into the high-resolution map and then mutated it into dengue NS1 sequence and refined the model using Coot and conducted real-space refinement using the Phenix software package to build the Dengue 2 NS1 dimer structure. In the figure 3, for the interpretation of the 8Å resolution loose tetrameric and hexameric maps, we fitted these maps using the high resolution structure of the NS1 dimer obtained from focused refinement of the one of the dimers of the loose tetramer.

We have included this in the method.

“Protein structure building

Building of the high resolution structures of sNS1 - a dimer of the loose tetramer (PDB: 7WUS) and the stable tetramer (PDB: 7WUT) structures, were done in this sequence – (1) approximate fitting using the Zika NS1 dimer structure (PDB 5GS6) as rigid bodies into the density using “fit-in-map” function in Chimera, (2) mutation of residues to that of dengue NS1 sequence using the program Coot and (3) then fitting the individual residues into densities. We then refined the structures using the “phenix.real_space_refine” procedure in the Phenix software package, with default parameters and rigid body refinement using secondary-structure and torsion angle restraints. The final coordinates containing an asymmetric unit (i.e., a protomer in the stable tetramer structure) were then used to build the biological unit (full tetramer) by using the command “sym” in UCSF Chimera. The final coordinates of the asymmetric units were checked using MolProbity. To interpret the 8Å resolution loose tetramer (PDB:7WUU) and hexamer (PDB:7WUV) maps, we fitted these maps by using the high resolution structure of the dimer from the loose tetramer (PDB: 7WUS). We then refined the structures using the “phenix.real_space_refine” procedure in the Phenix software package, with default parameters and rigid body refinement using secondary-structure and torsion angle restraints. Maps and structures shown in the figures were generated using UCSF Chimera and Coot.”

Besides, in the Extended Data Fig. 1A legend, it was written as: “(A) Crystal structure of dengue dimeric iNS1 (PDB:5GS6).” But the PDB: 5GS6 is actually a Zika virus NS1 model. Therefore, the authors should thoroughly clarify the models used in the manuscript, which is critical for a structures-based paper.

We apologize for our mistake and thank the reviewer for pointing this out. We have modified it and re-drawn this figure using the dengue dimeric iNS1 (PDB: 4O6B). Also see the figure below.

We have changed the caption from:

“(A) Crystal structure of dengue dimeric iNS1 (PDB:5GS6).”

To

“(A) Crystal structure of dengue dimeric iNS1 (PDB:4O6B).”

3. Line 457 (Data availability): several EM-maps of this study are reconstructed with low-resolutions (~8Å), such as the Loose tetramer (EMD: 32840 and PDB: 7WUS), hexamer (EMD: 32842 and PDB: 7WUV), and authors also built models for the low-res maps. It’s quite confusing when I am reading, and I would say no one can build models with such maps. Did the author really build models with the maps, or just fitted models in? If a model is not built with a map, it should have no PDB code. A more rigorous solution is: to consider just uploading the low-res maps without models.

Thank you for this comment. For the loose tetramer, we did build the structure of one dimer (PDB: 7WUS) of the loose tetramer because focused refinement on one dimer did provide us with a 3.4 Å map. To visualize the entire loose tetramer, we then fitted this high resolution dimer model structure into the 8Å map. This loose tetramer structure coordinates is then deposited into PDB without its side chains signalling the resolution of the map is low. For the hexamer, as the map is low resolution, we only conducted fitting of the same dimer models (PDB: 7WUS) into the map and also PDB coordinate was deposited without side chains. When submitting coordinates into the PDB, we did declare that they are 8Å resolution. Generally, the cryoEM field encourages deposition of fitted coordinates into the low resolution maps into PDB, because it will show the quaternary arrangements of molecules which is still an invaluable structural information.

We have clarified how we fitted the structures – see answer to reviewer #2, question 2.

4. A previous publication (Sci Transl Med 2019; PMID: 31243154) demonstrated the Dengue NS1 mutation T164S, which is located at the interface of NS1 tetramer, alters the disease severity in mice, and the NS1 hexamer was predicted to be less stable with this mutation. It is commended the authors can cite and discuss the role of this mutation with the “elongated β -sheet” discovery. It may extend the interest and improve the quality of the manuscript.

We thank the reviewer for this suggestion.

We have included it in the discussion:

“Previous analysis (Chan et al., 2019) , comparing two DENV2 strains that causes different disease severity, points to a mutation on NS1 (T164S) that leads to increased disease severity. This mutation also results in an increased in sNS1 production in mammalian cells. Analysis of this residue in our stable tetramer structure (Supplementary Fig.12) shows it is at the interacting interface to the hydrophilic N-terminal part of the elongated β -strand of the opposite dimer. This mutation may slightly increase the hydrophilicity of this interacting interface, thus encouraging interactions between two dimers. However, this will require further investigation.”

5. For the putative densities of lipids, basically, I suppose it is hard to state there are lipids with the maps, especially when they were refined with D2 or D3 symmetry.

We thank the reviewer for the comment. For the loose tetramer, we conducted focused refinement (with C1 symmetry) on one of its dimers that produced a high resolution map and we observed strong uninterpreted densities above the β -roll density. For the hexamer, there is sparse density in the core of the hexamer, and this map was made by imposing D3 symmetry. But the reviewer is right about difficulties in assigning lipid density. We have now removed speculations of their densities in the Fig 3.

Minor comments:

1. Please use correct and standard descriptions in virology. Line 15: “...significant human pathogens like West Nile (WNV), yellow fever (YF) and Zika virus (ZIKV)...” The pathogens are “West Nile virus” and “Yellow Fever virus”, but not “West Nile” and “yellow fever”. Yellow Fever (YF) is the name of the infectious disease, not the name of the pathogen. The sentence may be written as “significant human pathogens, such as West Nile virus (WNV), Yellow Fever virus (YFV) and Zika virus (ZIKV)”.

Thank the reviewer for this correction, we have modified it in the manuscript.

From

“which also includes significant human pathogens like West Nile(WNV), yellow fever(YF) and Zika virus (ZIKV).”

To

“which also includes significant human pathogens, such as West Nile virus(WNV), yellow fever virus (YF) and Zika virus (ZIKV).”

2. Line 427 (Protein structure building): just feel curious why the authors used a Zika NS1 model as the initial model, since there are several published Dengue NS1 models (e.g., 4O6B).

We use ZIKA NS1 structure as initial model because both dengue and zika NS1 have largely similar structures and the crystal structure of full-length ZIKV virus is more complete - DENV NS1 (PDB: 4O6B) lacks the flexible wing loop regions (Residues 108-128, and Residues 163-164).

Reviewer #3 (Remarks to the Author):

The Dengue virus non-structural protein 1 (NS1) is a multifunctional protein. Inside infected cells, NS1 is involved in viral replication, by associating with cellular membranes to form the viral replication complexes. Additionally, NS1 can also be secreted, and the amount of secreted NS1 has been linked with Dengue hemorrhagic fever. While there are structures available of intracellular NS1 (by X-ray crystallography), there are no structures for soluble NS1.

In this manuscript Shu et al study complexes of secreted dengue virus NS1 protein by cryo-EM. They are able to solve the structure of stable and loose tetramers, and of hexameric structures of NS1 (to a lower resolution). The authors also solve the structure of secreted NS1 in the presence of a Fab, which appears to only bind to loose tetramers or to hexamers; not to stable tetramers. Comparing the structures of the previously published intracellular tetramers with the stable and loose secreted tetramers, the authors found a key difference in the N-terminal domain conformation: it adopts an elongated beta-sheet in the stable tetramer; and a beta-roll in the loose tetramers and the hexamers.

While the manuscript is clearly written and the results are solid, I fail to understand how the structure of soluble NS1 helps understand its effect on Dengue hemorrhagic fever. The authors could address this issue by performing TEER in the presence of stable tetramers compared to loose tetramers. Additionally, the Fab used by the authors only partially prevents NS1 from inducing human umbilical vein endothelial cell hyperpermeability; but according to Fig 1E only ~23% of the particles are in a stable tetramer conformation. So the observation that the Fab cannot bind stable tetramers does not seem to explain the low effect seen by the Fab (I would expect more protection if the Fab was blocking the effect of ~77% of NS1).

We thank the reviewer for this question. It will be interesting to know the hyperpermeability activity difference between the loose and stable tetramer but unfortunately, we are unable to separate and purify them. This is because, they have the same chemical properties and molecular weight.

The mechanism of how sNS1 and which oligomerization states causes endothelial hyperpermeability is unknown and likely to be very complex. The results from the antibody:sNS1 TEER assay is a largely a qualitative assay, we are unsure if we can quantify how efficient is the stable tetramer in causing hyperpermeability.

Minor comments:

- Figure 1B requires a loading control of some sort; a sample incubated 30 min with a crosslinker should not have more monomer than when it is incubated only 10 min.

We thank the reviewer for this comment and we have repeated this experiment several times, they are largely similar. We have included one of them to replace that in figure 1A – see below.

- It is unclear what the authors are showing in the “Side view” panels of Fig 3Bii. Why is there so much unoccupied density? Or is it the other dimer represented as a surface?

We thank the reviewer for pointing it out. The neighboring dimers are represented as surfaces, they are not uninterpreted density.

We have now clarified this in the legend.

From:

Figure 3(B) Comparison of our sNS1 hexamer with the dengue iNS1 hexamer from the crystal packing. (i) Top and side views of the 8Å resolution sNS1 hexamer structure. Left: fit of three dimers into the cryoEM map. We are unable to discern whether sNS1 adopts a β -roll or an elongated β -sheet structure, as the interior of the hexamer contains unfeatured density. (ii) The previously published dengue iNS1 structure shows hexamers due to crystal packing; however, there is little interaction between the iNS1 dimers (left; top view). The dimers in the hexamer (right; side view) are also oriented differently to our hexameric structure (in (i), side view).

To

Figure 3 (B) Comparison of our sNS1 hexamer with the dengue iNS1 hexamer from the crystal packing. (i) Top and side views of the 8Å resolution sNS1 hexamer structure. Left: fit of three dimers into the cryoEM map. We are unable to discern whether sNS1 adopts a β -roll or an elongated β -sheet structure, as the interior of the hexamer contains unfeatured density. (ii) The previously published dengue iNS1 structure shows hexamers due to crystal packing; however, there is little interaction between the iNS1 dimers (left; top view). The dimers in the hexamer (right; side view) are also oriented differently to our hexameric structure (in (i), side view). For the side view, the protomers within one dimer are colored in pink and green ribbons, and the other dimers are shown as white surface representations.

- 1 Gutsche, I. *et al.* Secreted dengue virus nonstructural protein NS1 is an atypical barrel-shaped high-density lipoprotein. *Proc Natl Acad Sci U S A* **108**, 8003-8008, doi:10.1073/pnas.1017338108 (2011).
- 2 Benfrid, S. *et al.* Dengue virus NS1 protein conveys pro-inflammatory signals by docking onto high-density lipoproteins. *EMBO Rep*, e53600, doi:10.15252/embr.202153600 (2022).

- 3 Flamand, M. *et al.* Dengue virus type 1 nonstructural glycoprotein NS1 is secreted from mammalian cells as a soluble hexamer in a glycosylation-dependent fashion. *J Virol* **73**, 6104-6110, doi:10.1128/JVI.73.7.6104-6110.1999 (1999).
- 4 Muller, D. A. *et al.* Structure of the dengue virus glycoprotein non-structural protein 1 by electron microscopy and single-particle analysis. *J Gen Virol* **93**, 771-779, doi:10.1099/vir.0.039321-0 (2012).
- 5 Liang, B. *et al.* Secreted dengue virus NS1 is predominantly dimeric and in complex with high-density lipoprotein. *bioRxiv* (2022).
- 6 Glasner, D. R., Puerta-Guardo, H., Beatty, P. R. & Harris, E. The Good, the Bad, and the Shocking: The Multiple Roles of Dengue Virus Nonstructural Protein 1 in Protection and Pathogenesis. *Annu Rev Virol* **5**, 227-253, doi:10.1146/annurev-virology-101416-041848 (2018).

Reviewers' Comments:

Reviewer #1:

Remarks to the Author:

The authors convincingly show that tetrameric forms of NS1 exist in the context of recombinant expression and provide the first high-resolution structure of sNS1 as described in the initial manuscript. In their response to the initial review, they attempt to show that these oligomeric forms are present in the context of infected cells (re-analysis of published cryo-EM, chemical crosslinking). They did not directly address the central point of the review, which queried whether these tetramers have biological significance. If shown, this would have a broad impact in the field and the structures presented here would provide an invaluable structural framework.

There are several avenues to support the biological significance of the tetramer:

1. Demonstrate its "dominance" in native/native-like samples (clinical samples, supernatant of infected cells...). The crosslinking experiments attempt to support this but did not provide a strong support in their current form (cf. comment on Fig. 1A-B below)
2. Show that the stable tetrameric form is the functional form for at least one of NS1 functions (cf. point below about TEER assay of the purified stable tetramer)
3. Identify evidence that the tetrameric interface is important (conservation analysis, analysis of natural and site-directed mutants)

In the absence of at least one of these, the study is very interesting to the dengue virus field but possibly of limited interest to the broader scientific community since the tetramers may or may not be artefacts of recombinant expression.

Introduction

L. 48: "It is eventually transported to the cell membrane where some is secreted outside the cell, becoming sNS1."

Is there evidence that NS1 reaches the plasma membrane? Isn't release from the membrane thought to happen earlier?

L.75: "some of these structures are determined to $\sim 3.5\text{\AA}$ resolution."

It would be clearer to rephrase and state the 3 resolutions for the tetrameric reconstructions (or none).

L. 77-78: "Only a minority of the sNS1 population is hexameric and this cryoEM structure was determined to 8\AA resolution."

The authors should remind the reader that the recombinant NS1 is discussed here.

L81-82: "This is consistent with our results showing the ability of the Fab 5E3 to only partially inhibit sNS1-induced endothelial cell permeability."

This implies that the stable tetrameric form induces endothelial cell permeability, which has not been shown. Same for the concluding remarks l. 262-264 and 265-268. This should be explicitly stated if supported or rephrased. Cf. suggestions below to establish this point.

Results

"The recombinant sNS1 from DENV2 contains heterogenous oligomeric states with tetramers as the predominant population."

This section and the experiments that accompany it remain inconclusive. The section title states that

tetramers are “the predominant population”, which is inferred only from analysis of the cryo-EM classes. Unfortunately, this type of analysis is subject to several caveats as described below. An orthogonal technique is needed to support this conclusion. Many approaches exist and some are readily accessible such as AUC, SAXS, SEC-MALLS or native gels (perhaps mass photometry if available).

Moreover, experiments shown in Fig. 1A and 1B seem to show that the recombinant and infected cell-derived NS1 differ significantly - as detailed next - while the text in this section and in the conclusion gives the opposite impression.

L. 94: According to the labels (but it would be important to see the molecular weights to confirm), Fig.1B shows that, before cross-linking and despite the presence of SDS, there is equal amount or more hexamer than tetramer (bearing in mind that Western blot is semi-quantitative at best). Upon incubation with BS, the tetrameric species become dominant. However, the experiment suggests that BS is unable to crosslink the “native” hexamer and more importantly that the protein from infected cell supernatant differs from the recombinant NS1 in several ways (hexamer/tetramer present at $t=0$; no monomer; no clear evidence of crosslinking of the hexamer).

Fig. 1D: what oligomeric form does the butterfly 2D class represent (1st panel on the left)? How populated is this class?

Fig. 1E: This analysis provides a good indication that tetrameric forms are dominant in the particles used for 3D classification. However, issues with this type of quantification are that (1) it relies on what ends up in imaged areas of the grid with thin ice and (2) does not take into consideration particles that did not make it to 3D classification (~55% here or 1.4m particles). It is possible that some forms prefer the carbon surface of the grid or thicker ice (e.g. the hexamers). Labile oligomers may also be denatured/disassembled during the freezing process (e.g. at the air-water interface). These caveats (and the fact that the protein is recombinantly expressed) are not mentioned when discussing the hexameric structure later on:

“L.206-208: However, we observed loose tetramers forming ~74% of the particle population (Fig. 1D), while hexamers only form ~3%, suggesting that the loose tetramer may be a preferred oligomerization state.”

As mentioned above, this wouldn't be such a concern if an orthogonal method is used to confirm the conclusions.

3. Discussion:

Determination of whether the stable tetramer is a pathogenic form may not be as complex as described here. Since Fab 5E3 does not bind the stable tetramer, it seems straightforward to purify this form from all other sNS1 species using affinity depletion by 5E3 (immuno-precipitation or immuno-affinity column). It will then be possible to estimate in a TEER experiment what proportion of the specific activity of the unpurified sNS1 sample comes from the stable tetramer.

The new discussion of mutation T164S is interesting but incomplete. The following points should be clarified: (1) the region labeled N is not the true N-terminus of the protein, (2) Supp. Fig. 12 is a homology model (?) of the stable tetramer (i.e. not a real structure that could conceivably be totally different), (3) where is residue 164 in the loose tetramer and hexamer? Without this information, it is impossible to know whether the impact of the mutation may also be explained by an improved stability of the loose tetramer and/or hexamer.

Benfrid et al reported negative-stain EM not cryo-EM.

L. 346: "It was shown that sNS1 is one of the major factors contributing to the development of DHF/DSS in patients." Please provide the reference for this.

L. 347-348: As discussed above, it is possible that the "recombinantly expressed" protein does not have "native oligomerization states". I suggest removing "native" or demonstrating this point.

Other:

In the reporting summary, Fab 5E3 is described as 5E5 (validation).

Whether the samples are boiled or not should be described in the figure legend where appropriate. The presence of a monomeric species in Fig. 1A suggests that the sample at t=0 is boiled like the rest of the BS-crosslinked samples. In 1B however the t=0 sample is unlikely to have been boiled (no presence of monomer and tetramer/hexamer partially preserved).

Reviewer #2:

Remarks to the Author:

The authors have carefully addressed the questions. I have one more suggestion for the new version: The workflow for uncomplexed NS1 in Fig.S2 is not very clear. It suggests listing more details of the EM data processing workflow, e.g., which EM map is used to build which model. Another workflow for the Fab/NS1 dataset is also required.

Reviewer #1 (Remarks to the Author):

The authors convincingly show that tetrameric forms of NS1 exist in the context of recombinant expression and provide the first high-resolution structure of sNS1 as described in the initial manuscript. In their response to the initial review, they attempt to show that these oligomeric forms are present in the context of infected cells (re-analysis of published cryo-EM, chemical crosslinking). They did not directly address the central point of the review, which queried whether these tetramers have biological significance. If shown, this would have a broad impact in the field and the structures presented here would provide an invaluable structural framework.

We thank the reviewer for all the helpful comments and suggestions which make our manuscript substantially clearer.

There are several avenues to support the biological significance of the tetramer:

1. Demonstrate its “dominance” in native/native-like samples (clinical samples, supernatant of infected cells...).

Thank the reviewer for this suggestion.

Purifying sNS1 from blood or dengue infected cell culture for cryoEM analysis or biochemical characterization necessitates the use of antibodies to pull down these sNS1, as we have shown that the loose tetramer and hexamer structures are highly sensitive to antibody binding. The antibody will break them down into dimers, which could be the reason why all currently solved antibody:sNS1 structures are of Fab complexed with dimers¹⁻³. We have tested our panel of antibodies and showed that all caused reduced NS1 endothelial hyperpermeability activity (see figure below), this suggests that they altered the structure of the sNS1 and hence cannot be used for purification of sNS1 for structural determination.

There are multiple problems with purifying sNS1 from blood samples: (1) not getting enough protein for cryoEM studies, (2) the sNS1 complexes will be extremely heterogenous as they might have polyclonal antibodies bound to them, or (3) sNS1 is bound to lipid membranes and HDL etc. We therefore foresee that this would become an entirely new study which is beyond the scope of this paper.

The crosslinking experiments attempt to support this but did not provide a strong support in their current form (cf. comment on Fig. 1A-B below)

Regarding the crosslinking experiments, we found that the PEG8000 used for precipitating the sNS1 from TCS of infected cells affected the crosslinking results, removal of PEG8000 resolved the problem, for details please see answer to question 10.

2. Show that the stable tetrameric form is the functional form for at least one of NS1 functions (cf. point below about TEER assay of the purified stable tetramer)

Thank the reviewer for this suggestion.

As reviewer has suggested, we have tried to purify stable tetramer by removing all the other sNS1 oligomers (loose tetramers and hexamers) by using antibody 5E3 bound to protein G magnetic beads – we named this sample, stable tetramer concentrated sNS1 (STconc_sNS1). Less than 20% sNS1 was left in the flow-through after two rounds of purification, as determined by using UV spectrophotometry. We examined this sample under cryoEM and have done 2D and 3D classifications. After 3D classification (Table below), ~75% of the particles are stable tetramers, while ~25% are loose tetramers, no hexamers were detected (new supplementary Table 2B). This purification method successfully concentrated the stable tetramer although some loose tetramer was also present. Next we performed the TEER with the STconc_sNS1. Results showed that the TEER activity of STconc_sNS1 is similar to the sNS1 treated with Fab 5E3 – this suggests that the stable tetramer is functional. This is presented in the new main figure 4D.

Recombinant sNS1, particle distribution (Main figure 1E)

	Stable-Tetramer	Loose-Tetramer	Hexamer
Particles	278366	903489	37422
Percentage	22.8%	74.2%	3.1%

STconc_sNS1 particle distribution (New Supplementary Table 2B)

	Stable Tetramer	Loose Tetramer	Hexamer
Particles	315671	102939	0
Percentage	75.4%	24.6%	0 %

New main figure 4D

Fig. 4 (D) TEER assay showing Fab 5E3 can only partially prevent recombinant sNS1 from inducing human umbilical vein endothelial cell hyperpermeability. STconc_sNS1 (concentrated stable tetramer) has similar endothelial cell hyperpermeability activity as the sNS1+Fab 5E3 sample. Data are the mean \pm SEM from three independent experiments. Statistically significant differences between distinct groups (all time points combined for each

group) compared to the untreated groups were determined by a two-way ANOVA analysis using Dunnett's test for multiple comparisons. (**p < 0.001).

We have included this in the result:

“To further confirm that the stable tetramer is functional, we tried to purify the stable tetramer by harvesting the flow-through after the recombinant sNS1 sample has been exposed to the antibody 5E3:protein G magnetic beads. This sample is hereafter named, STconc_sNS1. The concentration of STconc_sNS1 sample is less than 20% of that of the original sNS1 protein. CryoEM 3D classification of the particles shows there are 75% stable tetramer and 25% loose tetramer (Supplementary Table 2B). This method successfully concentrated the stable tetramer, although some loose tetramers are present. The STconc_sNS1 sample also has similar endothelial hyperpermeability activity as when the sNS1 is complexed with Fab5E3 (Figure 4D). This suggests the stable tetramer has hyperpermeability activity.”

3. Identify evidence that the tetrameric interface is important (conservation analysis, analysis of natural and site-directed mutants)

Thank the reviewer for this suggestion.

We have done the conservation analysis and included this in the supplementary figures. This conservation analysis shows that the sequence of the elongated β -sheet is highly conserved amongst 130 flaviviruses - see the figure below. A paper on ZIKV NS1⁴ also shows the membrane-associated region (residue 1-30) is the most conserved in the NS1 sequences.

Figure above: The NS1 surface is colored according to sequence conservation from the most conserved (dark magenta) to the most divergent (dark cyan) based on an alignment of NS1 sequences from 130 flaviviruses using the ConSurf server⁵. The result shows that most region of the elongated β -sheet (dotted black lines) is conserved.

We have included this in the discussion:

“Sequence analysis and comparison of the flavivirus NS1 shows that the N-terminal domain (β -roll or elongated β -sheet) (residue 1-30) is highly conserved (Supplementary figure 4E), suggesting conservation of this region might be important for essential functions.”

We have looked through the literature on mutations at the interacting interface (β -roll or elongated β -sheet) between dimers. However, most mutations are characterized by how they would affect intracellular anti-viral responses and replication⁶⁻¹⁰ which might not require the secreted form of NS1 (sNS1) to be assembled. These papers show the N-terminal regions (β -roll or elongated β -sheet) mutations or insertion either abolished replication or virus plaque formation.

We found a publication¹¹ that show a mutation (residue 11) at the N-terminal end of the β -roll or elongated β -sheet can increase sNS1 secretion.

We have included this in the discussion:

“Another mutation that can also increase the secretion of sNS1 is located at the N-terminal end (residue 11) of either the elongated β -strand or β -roll.”

4. In the absence of at least one of these, the study is very interesting to the dengue virus field but possibly of limited interest to the broader scientific community since the tetramers may or may not be artefacts of recombinant expression.

Thank the reviewer for all these suggestions. For the native NS1, as indicated in our answer to question 1, currently there are still multiple problems with purifying native NS1 for structural analysis. It is very common that recombinant protein is used for structural studies because this is often the only way to obtain high quantity of pure samples.

Introduction

5. L. 48: “It is eventually transported to the cell membrane where some is secreted outside the cell, becoming sNS1.”

Is there evidence that NS1 reaches the plasma membrane?

We apologize for this confusion.

Previous work from Prof. Michael S Diamond group¹² have shown that NS1 expression on the cell surface was detected by flow cytometry using the anti-DENV NS1 antibody. We have included this reference.

Isn't release from the membrane thought to happen earlier?

In the NS1 lifecycle, after dimerization and glycosylation, some NS1 is transported to reside on the cell membrane, while others are secreted outside the cell. To make it clearer, we have modified it.

To

“iNS1 (45 kDa) is first made in the endoplasmic reticulum (ER) as a monomer. It forms dimers in the ER and then is glycosylated in the trans-Golgi network. Some NS1 is transported to the cell membrane, while others are secreted outside the cell, becoming the sNS1.”

6. L.75: “some of these structures are determined to ~3.5Å resolution.”

It would be clearer to rephrase and state the 3 resolutions for the tetrameric reconstructions (or none).

Thank you for pointing this out, we have modified it.

From

“some of these structures are determined to ~3.5Å resolution.”

To

“The stable tetramer structure is determined to ~3.5Å resolution while the overall resolution of the loose tetramer is 8 Å. We have also done focused refinement on one of the dimers of the loose tetramer and it is determined to 3.4Å resolution.”

7. L. 77-78: “Only a minority of the sNS1 population is hexameric and this cryoEM structure was determined to 8Å resolution.”

The authors should remind the reader that the recombinant NS1 is discussed here.

Thank you for pointing this out, we have modified it.

“Only a minority of the sNS1 population is hexameric and this cryoEM structure was determined to 8Å resolution.”

To

“Only a minority of the recombinant sNS1 population is hexameric and this cryoEM structure was determined to 8Å resolution.”

8. L81-82: “This is consistent with our results showing the ability of the Fab 5E3 to only partially inhibit sNS1-induced endothelial cell permeability.”

This implies that the stable tetrameric form induces endothelial cell permeability, which has not been shown. Same for the concluding remarks l. 262-264 and 265-268. This should be explicitly stated if supported or rephrased. Cf. suggestions below to establish this point.

Thank the reviewer for this suggestion.

Please see answers to question 2.

Results

“The recombinant sNS1 from DENV2 contains heterogenous oligomeric states with tetramers as the predominant population.”

9. This section and the experiments that accompany it remain inconclusive. The section title states that tetramers are “the predominant population”, which is inferred only from analysis of the cryo-EM classes. Unfortunately, this type of analysis is subject to several caveats as described below. An orthogonal technique is needed to support this conclusion. Many approaches exist and some are readily accessible such as AUC, SAXS, SEC-MALLS or native gels (perhaps mass photometry if available).

Thank the reviewer for making the above arguments for the need to have another orthogonal method to confirm the distribution for oligomeric form. We have now conducted AUC.

The AUC results show three peaks at molecular weights of 32, 146 and 251 kDa, corresponding to monomer, tetramer and hexamer. The monomer forms a minute population, there are some hexamers and the majority of the population is still tetramers.

We have included this in the results:

“Sedimentation velocity analytical ultracentrifugation (AUC) experiment of the recombinant sNS1 was conducted, and the molecular weights (MW) were derived by data analysis with SEDFIT based on sedimentation coefficient (Supplementary Figure 2). The estimated MW of the first peak is 32.5 Da, likely corresponds to a very minute population of monomeric particles. The MWs of the second and third peaks are 146 kDa and 251 kDa, and they correspond to tetrameric and hexameric sNS1 particles, respectively. The tetramers form the largest population of the sNS1 particles.”

Supplementary Figure 2: Sedimentation velocity analytical ultracentrifugation experiment of the recombinant sNS1. (A) Raw sedimentation profiles of absorbance at 280 nm versus particle radius for the recombinant sNS1. The sedimentation scans were coloured with the progressive rainbow colours according to the software default setting (B) Residual plot supplied by SEDFIT software showing the fitting goodness. (C) Continuous sedimentation coefficient distribution of the different sNS1 particle population – monomer, tetramers and hexamers with estimated MW of 32.5kDa, 146 kDa and 251 kDa, respectively.

10. Moreover, experiments shown in Fig. 1A and 1B seem to show that the recombinant and infected cell-derived NS1 differ significantly - as detailed next - while the text in this section and in the conclusion gives the opposite impression.

L. 94: According to the labels (but it would be important to see the molecular weights to confirm), Fig.1B shows that, before cross-linking and despite the presence of SDS, there is equal amount or more hexamer than tetramer (bearing in mind that Western blot is semi-quantitative at best). Upon incubation with BS, the tetrameric species become dominant. However, the experiment suggests that BS is unable to crosslink the “native” hexamer and more importantly that the protein from infected cell supernatant differs from the recombinant NS1 in several ways

(hexamer/tetramer present at t=0; no monomer; no clear evidence of crosslinking of the hexamer).

We thank the reviewer for pointing it out. It is interesting that there is no monomer in the t=0 sample, it seems to suggest that the dimer conformation is more stable in the infected cell-derived sNS1. We re-examine the differences in the purification of the recombinant sNS1 and the one from tissue culture supernatant (TCS) of infected cell for this western blot assay. For recombinant sNS1, we expressed the protein and then purify them through nickel column and then gel filtration. For the sNS1 from infected cells, we only concentrate sNS1 from TCS by PEG8000 precipitation before conducting the western blot experiment. One possibility is PEG8000 might stabilize sNS1 protein.

We then try to treat the sNS1(PEG8000 precipitated from infected cell TCS) by adding detergent for removal of lipid or by conducting chloroform partitioning to remove PEG8000¹³.

Above figures show various treatments to the sNS1 from PEG8000 precipitated from infected cell TCS (named: PEG_TCS_sNS1). (A) Addition of detergent (DDM or Triton X) largely did not change the profile of the bands compared to the untreated sample, (B) when PEG8000 is removed by chloroform partitioning, the hexamer band in the uncrosslinked PEG_TCS_sNS1 sample disappeared and the monomer band appeared. This indicates that PEG8000 was stabilizing the hexamers in the original PEG_TCS_sNS1 sample. There are double bands at both the dimer and tetramer bands suggests the NS1 from infected cells may contain different glycosylation states. When we crosslinked this sample (PEG removed) with BS³ crosslinker, bands corresponding to hexamers and aggregates appeared and also the tetramer band intensified. We have now replaced the main figure 1B with the new figure (B) – see below.

Fig. 1 Profile of the recombinant sNS1 and sNS1 from the supernatant of infected cells of DENV2 (PVP94/07) clinical strain. Western blot denaturing SDS-PAGE gel of sNS1 cross-linked with BS3 at different incubation times: (A) recombinant C-terminal His-tag sNS1 (detected using Anti-polyHistidine-Peroxidase antibody) and (B) the sNS1 from the supernatant of infected cells (detected using anti-NS1 antibody). For the recombinant sNS1, aggregates, hexamers, tetramers, dimers and monomers were seen. For the sNS1 of infected cells crosslinked sample, we observed aggregates, hexamers, tetramers and dimers. All samples were added the SDS-PAGE loading buffer, boiled for 5 min and then analyzed on a 4–20 % SDS-PAGE gradient gels (C) A cryoEM micrograph of the recombinant sNS1 showing distinct-shaped particles, e.g., butterfly-shaped (red circles) and cube-shaped particles (purple circles). Scale bar is 10 nm. (D) The 2D class averages indicate that the particles are heterogenous, with some likely existing as tetramers (stable and loose forms) and others as hexamers. The first panel represents the typical 2D average side view which is similar between all oligomeric states and the other three represent the top views of different oligomeric forms. (E) The distribution of particles in different oligomerization states determined after 3D classification and refinement. Most of the particles are tetramers (loose and stable) and hexamers are a minority in the population.”

We have written the treatment of PEG8000 precipitated sample by using chloroform partitioning before western blot analysis in the methods section- see below.

“Purification of sNS1 from dengue virus infected cell for Western Blot analysis

C6/36 cells were grown to ~80% confluency before infecting with DENV 2 PVP94/07 at a multiplicity of infection of 0.1. Infected cells were incubated at 29°C for 3 days. Tissue culture supernatant was clarified by centrifugation at 8,000g for 30 min. Virus were removed by precipitating with 8% polyethylene glycol 8,000 (PEG8000) and then centrifugation to spin down the virus particles. To precipitate sNS1 from the remaining supernatant, more PEG8000 was added to achieve a final concentration of 12% and the mixture was kept for overnight. After centrifugation, the pellet containing sNS1 was resuspended in phosphate buffer saline (PBS). To remove PEG8000 from the final sample, 200 µL of chloroform was added to 200 µL of the sNS1 sample and mixed for 10 mins. It was then centrifuged for 5 min at 10,000 g to allow separation of the chloroform layer from the mixture. The aqueous phase (upper layer) containing sNS1 was collected for western blot analysis”

11.Fig. 1D: what oligomeric form does the butterfly 2D class represent (1st panel on the left)? How populated is this class?

The butterfly 2D class represents the side view of the loose tetramer, hexamer sNS1 and stable tetramer. Below is the projections of generated from our 3D maps, we have boxed out all the butterfly side views.

The projections of generated from our 3D map

It is difficult to identify the different states from the side view, so only one typical side view was shown in the figure 1 in the main manuscript. To make it clearer, we have modified the legend of figure 1. Also see below.

Legend of figure 1

Fig. 1 Profile of the recombinant sNS1 and sNS1 from the supernatant of infected cells of DENV2 (PVP94/07) clinical strain. Western blot denaturing SDS-PAGE gel of sNS1 cross-linked with BS3 at different incubation times: (A) recombinant C-terminal His-tag sNS1 (detected using Anti-polyHistidine–Peroxidase antibody) and (B) the sNS1 from the supernatant of infected cells (detected using anti-NS1 antibody). For the recombinant sNS1, aggregates, hexamers, tetramers, dimers and monomers were seen. For the sNS1 of infected cells crosslinked sample, we observed aggregates, hexamers, tetramers and dimers. All samples were added the SDS-PAGE loading buffer, boiled for 5 min and then analyzed on a 4–20 % SDS-PAGE gradient gels (C) A cryoEM micrograph of the recombinant sNS1 showing distinct-shaped particles, e.g., butterfly-shaped (red circles) and cube-shaped particles (purple circles). Scale bar is 10 nm. (D) The 2D class averages indicate that the particles are heterogenous, with some likely existing as tetramers (stable and loose forms) and others as hexamers. **The first panel represents the typical 2D average side view which is similar between all oligomeric states and the other three represent the top views of different oligomeric forms.** (E) The distribution of particles in different oligomerization states determined after 3D classification and refinement. Most of the particles are tetramers (loose and stable) and hexamers are a minority in the population.”

12. Fig. 1E: This analysis provides a good indication that tetrameric forms are dominant in the particles used for 3D classification. However, issues with this type of quantification are that (1) it relies on what ends up in imaged areas of the grid with thin ice and (2) does not take into consideration particles that did not make it to 3D classification (~55% here or 1.4m particles). It is possible that some forms prefer the carbon surface of the grid or thicker ice (e.g. the hexamers). Labile oligomers may also be denatured/disassembled during the freezing process (e.g. at the air-water interface). These caveats (and the fact that the protein is recombinantly expressed) are not mentioned when discussing the hexameric structure later on: "L.206-208: However, we observed loose tetramers forming ~74% of the particle population (Fig. 1D), while hexamers only form ~3%, suggesting that the loose tetramer may be a preferred oligomerization state."

As mentioned above, this wouldn't be such a concern if an orthogonal method is used to confirm the conclusions.

Thank the reviewer for this suggestion, we have conducted AUC.

Please see answers to question 9

13. Discussion:

Determination of whether the stable tetramer is a pathogenic form may not be as complex as described here. Since Fab 5E3 does not bind the stable tetramer, it seems straightforward to purify this form from all other sNS1 species using affinity depletion by 5E3 (immuno-precipitation or immuno-affinity column). It will then be possible to estimate in a TEER experiment what proportion of the specific activity of the unpurified sNS1 sample comes from the stable tetramer.

Thank the reviewer for this suggestion.

Please see answer to question 2.

14. The new discussion of mutation T164S is interesting but incomplete. The following points should be clarified: (1) the region labeled N is not the true N-terminus of the protein, (2) Supp. Fig. 12 is a homology model (?) of the stable tetramer (i.e. not a real structure that could conceivably be totally different), (3) where is residue 164 in the loose tetramer and hexamer? Without this information, it is impossible to know whether the impact of the mutation may also be explained by an improved stability of the loose tetramer and/or hexamer.

We thank the reviewer for this suggestion. We have modified this in the discussion.

- (1) We deleted the "N" in the figure.
- (2) We now mention in the legends that the molecule with the mutated residue is a homology model
- (3) We added panels to show the residue 164 in the loose tetramer and hexamer structures.

Supplementary Figure 13. (A) Zoom-in view of regions around residue 164. This residue lies in close proximity (Ca-Ca backbone distance is 5.8Å) to the highly hydrophilic elongated β -sheet of the opposite dimer. (B) A homology model of stable tetramer with S164 mutation showed that the mutation might increase the hydrophilicity of that region leading to increased interactions with the opposite dimer. (C) black Five-pointed stars indicates where the T164 are in different oligomers of the sNS1

We also added two lines in the discussion:

“Examination of this mutation on the other oligomerization states (loose tetramer and hexamer), shows it is located in the “greasy finger” adjacent to the β -roll. Since this region influence NS1 dimers interactions to each other or to lipid cargo, they might also affect the assemers.”

15.Benfrid et al reported negative-stain EM not cryo-EM.

Thank you for pointing this out, we have modified it.

“Benfrid et al., 2022 reported cryoEM 2D class averages of the purified sNS1:HDL complex containing heterogeneous particles with either two, three and four NS1 dimers on the HDL surface.”

to

“Benfrid et al., 2022 reported negative-stain EM 2D class averages of the purified sNS1:HDL complex containing heterogeneous particles with either two, three and four NS1 dimers on the HDL surface.”

16.L. 346: “It was shown that sNS1 is one of the major factors contributing to the development of DHF/DSS in patients.” Please provide the reference for this.

Thank the reviewer for pointing it out, we have now included the reference.

Ref¹⁴: “Avirutnan, P. et al. Vascular leakage in severe dengue virus infections: a potential role for the nonstructural viral protein NS1 and complement. J Infect Dis 193, 1078-1088, doi:10.1086/500949 (2006).”

17.L. 347-348: As discussed above, it is possible that the “recombinantly expressed” protein does not have “native oligomerization states”. I suggest removing “native” or demonstrating this point.

Thank you for pointing this out, we have removed it

“the quaternary organization of the different native oligomerization states”

to

“the quaternary organization of the different oligomerization states”

Other:

18.In the reporting summary, Fab 5E3 is described as 5E5 (validation).

Thank the reviewer for pointing it out. We have corrected it.

19.Whether the samples are boiled or not should be described in the figure legend where appropriate. The presence of a monomeric species in Fig. 1A suggests that the sample at t=0 is boiled like the rest of the BS-crosslinked samples. In 1B however the t=0 sample is unlikely to have been boiled (no presence of monomer and tetramer/hexamer partially preserved).

Thank you for pointing this out. We have added this in the legend. Also see answer to question 10.

We boiled all the samples including the t=0 sample- see below.

Fig. 1 Profile of the recombinant sNS1 and sNS1 from the supernatant of infected cells of DENV2 (PVP94/07) clinical strain. Western blot denaturing SDS-PAGE gel of sNS1 cross-linked with BS3 at different incubation times: (A) recombinant C-terminal His-tag sNS1 (detected using Anti-polyHistidine–Peroxidase antibody) and (B) the sNS1 from the supernatant of infected cells (detected using anti-NS1 antibody). For the recombinant sNS1, aggregates, hexamers, tetramers, dimers and monomers were seen. For the sNS1 of infected cells crosslinked sample, we observed aggregates, hexamers, tetramers and dimers. **All samples were added the SDS-PAGE loading buffer, boiled for 5 min and then analyzed on a 4–20 % SDS-PAGE gradient gels.** (C) A cryoEM micrograph of the recombinant sNS1 showing distinct-shaped particles, e.g., butterfly-shaped (red circles) and cube-shaped particles (purple circles). Scale bar is 10 nm. (D) The 2D class averages indicate that the particles are heterogenous, with some likely existing as tetramers (stable and loose forms) and others as hexamers. **The first panel represents the typical 2D average side view which is similar between all oligomeric states and the other three represent the top views of different oligomeric forms.** (E) The distribution of particles in different oligomerization states determined after 3D classification and refinement. Most of the particles are tetramers (loose and stable) and hexamers are a minority in the population.”

Reviewer #2 (Remarks to the Author):

The authors have carefully addressed the questions. I have one more suggestion for the new version:

The workflow for uncomplexed NS1 in Fig.S2 is not very clear. It suggests listing more details of the EM data processing workflow, e.g., which EM map is used to build which model.

Another workflow for the Fab/NS1 dataset is also required.

Thank the reviewer for this suggestion. we have modified the Fig.S2 (Fig.S3 in new version) and add a workflow for Fab/NS1. Also see as below.

- 1 Biering, S. B. *et al.* Structural basis for antibody inhibition of flavivirus NS1-triggered endothelial dysfunction. *Science* **371**, 194-200, doi:10.1126/science.abc0476 (2021).
- 2 Modhiran, N. *et al.* A broadly protective antibody that targets the flavivirus NS1 protein. *Science* **371**, 190-194, doi:10.1126/science.abb9425 (2021).
- 3 Edeling, M. A., Diamond, M. S. & Fremont, D. H. Structural basis of Flavivirus NS1 assembly and antibody recognition. *Proc Natl Acad Sci U S A* **111**, 4285-4290, doi:10.1073/pnas.1322036111 (2014).
- 4 Baez, C. F. *et al.* Analysis of worldwide sequence mutations in Zika virus proteins E, NS1, NS3 and NS5 from a structural point of view. *Mol Biosyst* **13**, 122-131, doi:10.1039/c6mb00645k (2016).
- 5 Ashkenazy, H. *et al.* ConSurf 2016: an improved methodology to estimate and visualize evolutionary conservation in macromolecules. *Nucleic Acids Res* **44**, W344-350, doi:10.1093/nar/gkw408 (2016).
- 6 Choy, M. M. *et al.* A Non-structural 1 Protein G53D Substitution Attenuates a Clinically Tested Live Dengue Vaccine. *Cell Rep* **31**, 107617, doi:10.1016/j.celrep.2020.107617 (2020).
- 7 Hall, R. A. *et al.* Loss of dimerisation of the nonstructural protein NS1 of Kunjin virus delays viral replication and reduces virulence in mice, but still allows secretion of NS1. *Virology* **264**, 66-75, doi:10.1006/viro.1999.9956 (1999).
- 8 Blaney, J. E., Jr., Manipon, G. G., Murphy, B. R. & Whitehead, S. S. Temperature sensitive mutations in the genes encoding the NS1, NS2A, NS3, and NS5 nonstructural proteins of dengue virus type 4 restrict replication in the brains of mice. *Arch Virol* **148**, 999-1006, doi:10.1007/s00705-003-0007-y (2003).
- 9 Giraldo, M. I. *et al.* K48-linked polyubiquitination of dengue virus NS1 protein inhibits its interaction with the viral partner NS4B. *Virus Res* **246**, 1-11, doi:10.1016/j.virusres.2017.12.013 (2018).
- 10 Youn, S. *et al.* Evidence for a genetic and physical interaction between nonstructural proteins NS1 and NS4B that modulates replication of West Nile virus. *J Virol* **86**, 7360-7371, doi:10.1128/JVI.00157-12 (2012).
- 11 Ghosh, A. *et al.* Non-structural protein 1 (NS1) variants from dengue virus clinical samples revealed mutations that influence NS1 production and secretion. *Eur J Clin Microbiol Infect Dis* **41**, 803-814, doi:10.1007/s10096-022-04441-4 (2022).
- 12 Avirutnan, P. *et al.* Secreted NS1 of dengue virus attaches to the surface of cells via interactions with heparan sulfate and chondroitin sulfate E. *PLoS Pathog* **3**, e183, doi:10.1371/journal.ppat.0030183 (2007).
- 13 Bourdin, G. *et al.* Amplification and purification of T4-like escherichia coli phages for phage therapy: from laboratory to pilot scale. *Appl Environ Microbiol* **80**, 1469-1476, doi:10.1128/AEM.03357-13 (2014).
- 14 Avirutnan, P. *et al.* Vascular leakage in severe dengue virus infections: a potential role for the nonstructural viral protein NS1 and complement. *J Infect Dis* **193**, 1078-1088, doi:10.1086/500949 (2006).

Reviewers' Comments:

Reviewer #1:

Remarks to the Author:

The authors have added experiments that significantly improve the impact of the manuscript. We commend them for a study that identifies the tetrameric form of NS1 as a dominant species at least in recombinant expression and as a functional form enhancing permeability.

The crosslinking experiments show intriguing differences between recombinant expression and NS1 from infected cell supernatants. This does not need to be resolved here. I'm sure the origin and significance of these differences will be clarified in subsequent studies.